# The denitrosylase SCoR2 controls cardioprotective metabolic reprogramming

**Zachary W Grimmett[1,2], Rongli Zhang[2,3], Hua-Lin Zhou[2†], Qiuying Chen[4], Dawson Miller[4], Zhaoxia Qian[2], Justin Lin[2], Riti Kalra[2], Steven S Gross[4‡], Walter J Koch[5,6], Richard T Premont[2,7], Jonathan S Stamler[2,7]***

[1]Medical Scientist Training Program, Case Western Reserve University School of Medicine, Cleveland, United States; [2]Institute for Transformative Molecular Medicine, Department of Medicine, Case Western Reserve University School of Medicine, Cleveland, United States; [3]Cardiovascular Research Institute, Case Western Reserve University School of Medicine, Cleveland, United States; [4]Department of Pharmacology, Weill Cornell Medicine, New York, United States; [5]Department of Surgery, Duke University School of Medicine, Durham, United States; [6]Department of Medicine, Duke University School of Medicine, Durham, United States; [7]Harrington Discovery Institute, University Hospitals Cleveland Medical Center, Cleveland, United States

**\*For correspondence:**
jss156@case.edu

**Present address:** [†]Department of Biochemistry and Molecular Biology, Medical College of Georgia, Augusta University, Augusta, United States

[‡]Deceased

## eLife Assessment

This study provides new and interesting findings that SCoR2 acts as a denitrosylase to control cardioprotective metabolic reprogramming and prevent injury following ischemia/reperfusion. The **compelling** evidence is supported by a novel multi-omics approach, but questions remain regarding the stability and human relevance of BDH1 as well as the sufficiency of SCoR2. Overall, the work will be of interest to cardiovascular researchers and provides **valuable** information to the field, though some mechanistic aspects require further clarification.

**Abstract** Acute myocardial infarction (MI) is a leading cause of morbidity and mortality, and therapeutic options remain limited. Endogenously generated nitric oxide (NO) is highly cardioprotective, but protection is not replicated by nitroso-vasodilators (e.g., nitrates, nitroprusside) used in clinical practice, highlighting specificity in NO-based signaling and untapped therapeutic potential. Signaling by NO is mediated largely by *S*-nitrosylation, entailing specific enzymes that form and degrade *S*-nitrosothiols in proteins (SNO-proteins), termed nitrosylases and denitrosylases, respectively. SNO-CoA Reductase 2 (SCoR2; product of the *Akr1a1* gene) is a recently discovered protein denitrosylase. Genetic variants in SCoR2 have been associated with cardiovascular disease, but its function is unknown. Here, we show that mice lacking SCoR2/AKR1A1 exhibit robust protection in an animal model of MI. SCoR2 regulates ketolytic energy availability, antioxidant levels, and polyol homeostasis via *S*-nitrosylation of key metabolic effectors. Human cardiomyopathy shows reduced SCoR2 expression and an *S*-nitrosylation signature of metabolic reprogramming, mirroring SCoR2$^{-/-}$ mice. Deletion of SCoR2 thus coordinately reprograms multiple metabolic pathways—ketone body utilization, glycolysis, pentose phosphate shunt, and polyol metabolism—to limit infarct size, establishing SCoR2 as a novel regulator in the injured myocardium and a potential drug target.

## Introduction

Cardiovascular disease resulting in myocardial infarction (MI) is the leading cause of death in the U.S. (*Tsao et al., 2023*). In the setting of an acute MI, preservation of even an additional 5% of myocardial tissue can reduce mortality and progression to heart failure (*Stone et al., 2016*). However, despite decades of research, no FDA-approved drug can reduce infarct size, and post-MI morbidity/mortality remains high (*Johansson et al., 2017*). Targeting well-described mediators of myocardial ischemia–reperfusion (I/R) injury, such as intracellular $Ca^{2+}$ overload, pH imbalance, oxidative stress, or inflammation, has thus far been unsuccessful in mitigating lethal cardiac injury in humans (*Hausenloy and Yellon, 2013*; *Heusch, 2017*; *Lefer and Marbán, 2017*).

An extensive literature spanning over three decades details the protective effects of nitric oxide (NO) in myocardial ischemia (*Bolli, 2001*; *Schulz et al., 2004*; *Janssens et al., 2004*; *Lima et al., 2009*; *Gonzalez et al., 2008*; *Dezfulian et al., 2009*; *Duranski et al., 2005*; *Sun et al., 2007*), including regulation of mitochondrial energetics, $Ca^{2+}$ transport, HIF-1α-mediated hypoxia response, and cell death. It is increasingly appreciated that NO signals largely through protein *S*-nitrosylation (*Seth et al., 2018*), a ubiquitous post-translational modification that regulates protein activity with manifold functional consequences. *S*-nitrosylation has been convincingly demonstrated to exert a broadly protective role in myocardial I/R injury (*Lima et al., 2009*; *Sun et al., 2007*). Accumulating evidence implicates specific enzymes that attach (nitrosylases) and remove (denitrosylases) NO groups at target Cys residues (*Seth et al., 2018*; *Zhou et al., 2023a*; *Stomberski et al., 2022*), by direct analogy to kinases and phosphatases that add and remove phosphate groups (i.e., phosphorylation). Studies of the founding member of the denitrosylase family, *S*-nitrosoglutathione (GSNO) reductase (GSNOR), have demonstrated important roles in I/R and transverse aortic constriction models of cardiac injury in mice, and in cardiac arrest in humans (*Hayashida et al., 2019*). Targets of GSNOR include SNO-HIF-1α (*Lima et al., 2009*), SNO-ANT1 (*Tang et al., 2023*), and SNO-GRK2 (*Whalen et al., 2007*; *Huang et al., 2013*), suggesting GSNOR confers protection through multiple synergistic mechanisms.

SCoR2 (also known as AKR1A1) is a newly identified protein denitrosylase with distinct cellular distribution, substrates, and purview (*Anand et al., 2014*; *Stomberski et al., 2019*); it is mechanistically and functionally homologous to yeast SCoR1/Adh6 (*Anand et al., 2014*). SCoR2 is highly expressed in kidney proximal tubules, but also found in other tissues including the heart (*Zhou et al., 2019*). Inhibition of SCoR2 by genetic knockout or drug treatment protects mice from acute kidney injury (*Zhou et al., 2019*; *Zhou et al., 2023b*) and lowers cholesterol by inhibiting liver PCSK9 secretion (*Stomberski et al., 2022*). SCoR2 acts on seemingly distinct sets of SNO-protein substrates in the kidney and liver, suggesting that its broader role in biology remains to be understood. Genetic variants in SCoR2/*AKR1A1* are associated with heart failure (*Watanabe, 2019a*; *Watanabe et al., 2019b*; *Aragam et al., 2019*) and cardiovascular function in humans (*Cardiovascular Disease Knowledge Portal, 2022*; *Costanzo et al., 2023*); however, the role and function of SCoR2 in the cardiovascular system remains unexplored. We therefore assessed cardiac protection in mice lacking SCoR2 and employed a multi-omic approach to define SCoR2-dependent protective changes following cardiac ischemia/reperfusion injury.

## Results

### Cardiovascular physiology of SCoR2

Measures of cardiac function by echocardiography were unchanged at baseline in naive SCoR2$^{-/-}$ mice (identical to previously described global *Akr1a1*$^{-/-}$ mice; *Zhou et al., 2019*) relative to SCoR2$^{+/+}$, including ejection fraction, fractional shortening, left ventricular internal diameter at end-systole (LVIDs), and ventricular pressures (*Figure 1—figure supplement 1A–F*). SCoR2$^{-/-}$ mice also did not differ in blood pressure (*Figure 1—figure supplement 1G, H*), heart rate (*Figure 1—figure supplement 1I*), body weight (*Figure 1—figure supplement 1J*), or heart/lung weight (*Figure 1—figure supplement 1K*). Thus, knockout of SCoR2 has no major effect on baseline cardiac function.

### SCoR2$^{-/-}$-mediated cardioprotection in an animal model of MI

We employed a model of acute MI, in which the mouse left anterior descending (LAD) artery is ligated for 30 min, followed by 1–24 hr of reperfusion. LAD ligation induced ST-elevation, confirmed by electrocardiography (ECG), characteristic of acute myocardial ischemia. Mice lacking SCoR2 exhibited

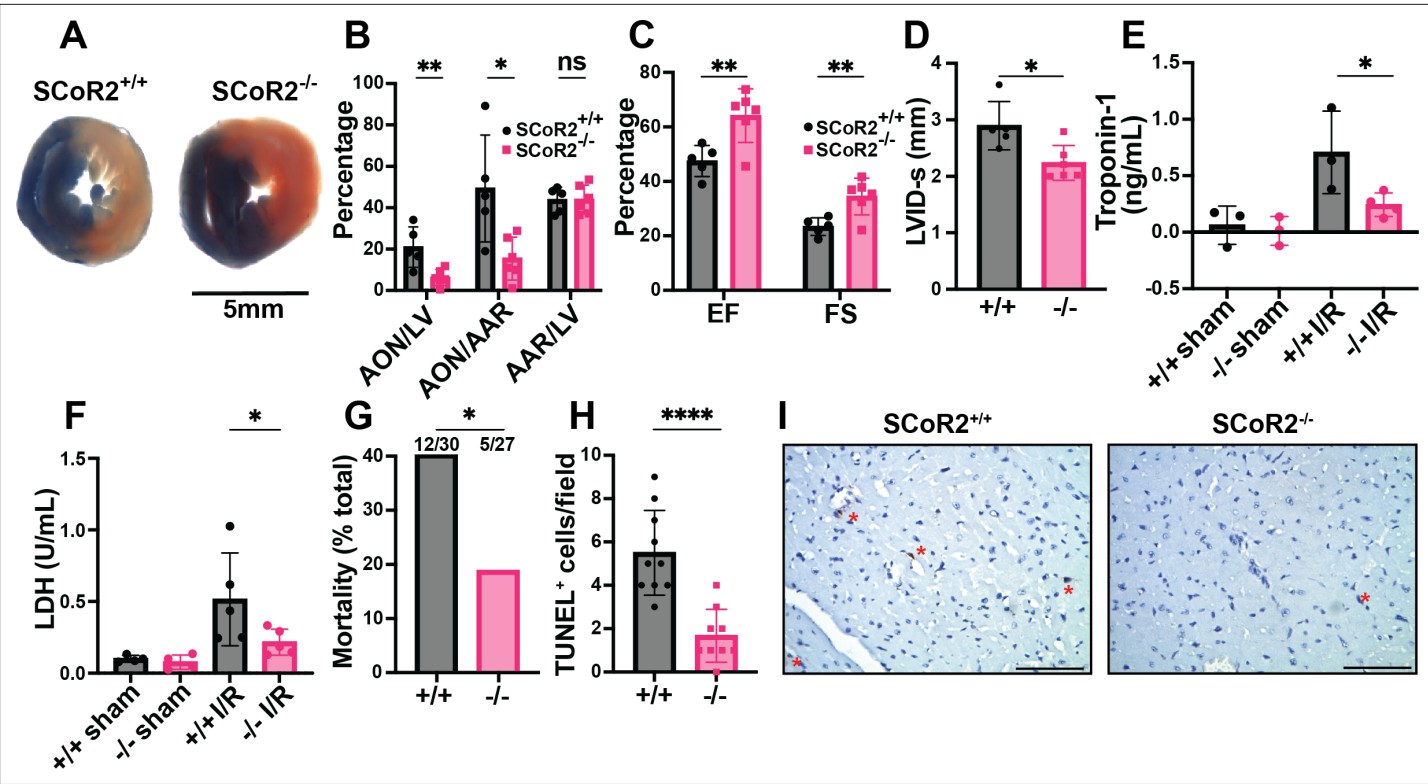

**Figure 1.** Knockout of SCoR2 protects from myocardial injury. (**A**) Representative myocardial infarct staining after ischemia–reperfusion (I/R) in SCoR2$^{+/+}$ (+/+) and SCoR2$^{-/-}$ (-/-) mice, taken from the same anatomical plane. Infarcted necrotic tissue is white, the area at risk is red, and tissue with normal perfusion is dark blue. (**B**) Quantification of the myocardial infarct size after I/R injury (24 hr reperfusion). AON (area of necrosis) is expressed as a percentage of the LV (left ventricle) and AAR (area at risk). (**C**) Quantification of left ventricular function after I/R injury (24 hr reperfusion). Ejection fraction (EF) and fractional shortening (FS) were determined by echocardiography. (**D**) Left ventricular internal diameter at end systole (LVID-s) was measured in SCoR2$^{+/+}$ and SCoR2$^{-/-}$ mice after I/R injury (24 hr reperfusion). (**A–D**) $N$ = 5 +/+, 6 -/- mice. (**E**) Serum troponin-1 concentration in +/+ and -/- mice after I/R injury (4 hr reperfusion); $N$ = 3–4 mice/condition. (**F**) Serum lactate dehydrogenase (LDH) concentration in +/+ and -/- mice after I/R injury (4 hr reperfusion) normalized to +/+ sham; N=4–5 mice/condition. (**G**) Quantification of post-MI survival at 4 hr post-reperfusion in -/- ($N$ = 27) and +/+ ($N$ = 30) mice. 12/30 +/+ mice and 5/27 -/- mice did not survive at 4 hr post-reperfusion. (**H, I**) Quantification of TUNEL$^+$ (apoptotic) nuclei in post-MI +/+ versus -/- myocardium at 4 hr post-reperfusion ($N$ = 3 each), with representative images in (**I**). Red asterisks indicate TUNEL$^+$ nuclei; scale bar = 50 μm. Significance in (**B–D**) assessed by two-tailed Student's $t$-test, in (**E**) by one-tailed Mann–Whitney test, in (**F**) by one-tailed Student's $t$-test, in (**G**) by one-sided chi-squared test, and in (**H**) by two-tailed Student's $t$-test; *p < 0.05, **p < 0.01, ****p<0.0001, ns = not significant.

The online version of this article includes the following figure supplement(s) for figure 1:

**Figure supplement 1.** Characterization of SCoR2$^{-/-}$ mice at baseline and 24 hr post-MI.

markedly reduced infarct size after I/R injury, as measured by differential Evans blue/TTC staining of left ventricle (LV) cross-sections distal to the ligation (***Figure 1A***). Area of necrosis as a percentage of total LV area and as a percentage of area at risk (AAR) was also decreased in SCoR2$^{-/-}$ mice relative to SCoR2$^{+/+}$ mice, though AAR/LV was not different (***Figure 1B***). Thus, SCoR2 deletion is protective in I/R injury, implying that its denitrosylated substrates contribute to tissue damage.

Cardiac function was assessed by transthoracic echocardiography 24 hr after I/R injury. Ejection fraction (EF) and fractional shortening (FS) were both better maintained in SCoR2$^{-/-}$ mice relative to SCoR2$^{+/+}$ (***Figure 1C***). LVID-s was reduced in SCoR2$^{-/-}$ hearts versus SCoR2$^{+/+}$ (***Figure 1D***), consistent with preserved LV function. Heart rate, stroke volume, and cardiac output remained unchanged in SCoR2$^{-/-}$ mice relative to SCoR2$^{+/+}$ (***Figure 1—figure supplement 1L–N***), and markers of cardiac hypertrophy (LV posterior wall (LVPW) and interventricular septum (IVS) thickness) were also unchanged (***Figure 1—figure supplement 1O–R***). Cardiac troponin-1 released into the bloodstream at 4 hr after LAD ligation was reduced by SCoR2 deletion (***Figure 1E***). Likewise, lactate dehydrogenase (LDH) levels in plasma were lower in injured SCoR2$^{-/-}$ mice compared to SCoR2$^{+/+}$ at 4 hr post-reperfusion (***Figure 1F***). Notably, SCoR2$^{-/-}$ mice demonstrated improved post-MI survival (12/30; 40%) relative to

SCoR2[+/+] mice (5/27; 19%) at 4 hr post-reperfusion, supportive of an acute cardioprotective pheno-type (*Figure 1G*). Fewer TUNEL[+] nuclei were observed in the myocardium of SCoR2[-/-] mice at 4 hr post-MI, indicating reduced apoptotic cell death (*Figure 1H, I*).

## SCoR2 mediates denitrosylation of cardiac SNO-proteins

The canonical NO-stimulated cGMP/protein kinase G pathway protects against cardiac injury (*Kukreja et al., 2012*), but cGMP levels after I/R were no different in SCoR2[+/+] versus SCoR2[-/-] mice (*Figure 2A*), suggesting protection mediated by protein *S*-nitrosylation (*Zhou et al., 2019*; *Stomberski et al., 2022*; *Figure 2—figure supplement 1A*). Indeed, while SCoR2 expression was verified in WT heart (*Figure 2—figure supplement 1B*), its activity was eliminated in hearts from SCoR2[-/-] mice (*Figure 2—figure supplement 1C*) and protein *S*-nitrosylation was elevated at baseline (i.e., in the uninjured SCoR2[-/-] mouse heart) (*Figure 2—figure supplement 1D, E*). Further, in SCoR2[+/+] heart lysates at baseline, addition of SNO-CoA (SCoR2 substrate and NO donor) (*Zhou et al., 2023a*) alone increased protein *S*-nitrosylation (*Figure 2B*), whereas co-administration of NADPH (the electron donor used by SCoR2 during denitrosylation) led to a decrease in cardiac SNO-proteins (*Figure 2C, D*). Although other denitrosylase enzymes are present in the heart, SCoR2 is responsible for approximately 65% of NADPH-dependent denitrosylase activity (*Figure 2—figure supplement 1C*); thus, SCoR2 regulates protein *S*-nitrosylation in the heart. Conversely, while SCoR2/AKR1A1 is required for vitamin C synthesis (*Lai et al., 2017*), mice were supplemented with vitamin C, and other putative carbohydrate substrates for SCoR2's aldo-keto reductase activity were undetectable (*Supplementary file 2*).

## Multi-omic analysis identifies SCoR2-regulated cardioprotective pathways

To identify proteins whose *S*-nitrosylation was regulated by SCoR2 following cardiac injury, heart lysates from SCoR2[+/+] versus SCoR2[-/-] mice at 4 hr post-I/R (vs. sham) were subjected to *S*-nitrosothiol resin assisted capture (SNORAC) and visualized by staining with Coomassie blue (*Figure 3A*). To elucidate mechanisms of SCoR2 deficiency-dependent cardioprotection and identify sites of regulation, we subjected post-I/R SCoR2[+/+] and SCoR2[-/-] mouse hearts to three complementary unbiased screens (done in triplicate): (1) SCoR2-dependent nitrosoproteome; (2) SCoR2 interactome; and (3) SCoR2-dependent metabolome (in heart and plasma) (*Figure 3B*).

LC–MS/MS of SNORAC samples from 4 hr post-I/R SCoR2[+/+] and SCoR2[-/-] mouse heart identified nearly 2000 unique SNO-proteins. Of these, 158 proteins showed increased *S*-nitrosylation (>1.2-fold) in post-I/R SCoR2[-/-] hearts relative to post-I/R SCoR2[+/+], comprising the SCoR2-dependent nitrosoproteome (*Figure 3B*, *Supplementary file 1*). Notably, metabolic proteins were heavily represented in the cardiac SCoR2-dependent nitrosoproteome (35/158 proteins; *Supplementary file 1*), similar to wild-type hearts (*Lau et al., 2021*).

Co-immunoprecipitation using anti-SCoR2 antibody in untreated SCoR2[+/+] heart lysate coupled with LC–MS/MS identified 515 SCoR2-associated proteins (SCoR2-dependent interactome) (*Figure 3B*; *Supplementary file 1*), of which 31 overlapped the SCoR2-dependent nitrosoproteome (*Figure 3B, C*). Since these 31 SNO-proteins both associated with SCoR2 and displayed SCoR2-dependent *S*-nitrosylation, they represent candidate SCoR2 denitrosylation substrates.

Untargeted metabolite profiling (using an LC/MS metabolomic platform) was performed in parallel using heart lysate and blood plasma from 4 hr post-I/R SCoR2[+/+] and SCoR2[-/-] mice to identify metabolites regulated by SCoR2-dependent SNO-proteins (SCoR2-dependent metabolome) (*Supplementary file 2*). These metabolite studies identified three metabolic pathways that were significantly perturbed by SCoR2 deletion: ketone body oxidation, pentose phosphate pathway (PPP), and polyol metabolism.

Finally, multi-omic analysis by integration of all this data identified two specific SCoR2-substrate SNO-proteins, β-hydroxybutyrate dehydrogenase (BDH1) and pyruvate kinase M2 (PKM2) (*Figure 3B, C*). This unbiased analysis suggests that these two enzymes are likely to be key contributors to metabolic reprogramming after cardiac I/R, identifying specific targets for SCoR2 in its regulation of cardiac metabolism.

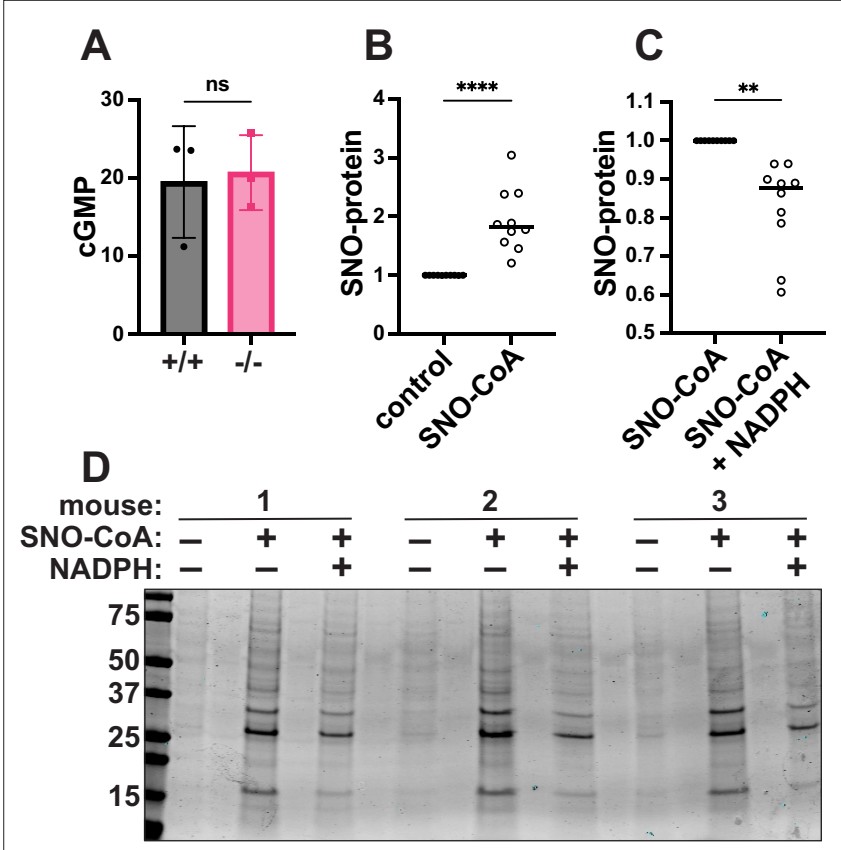

**Figure 2.** SCoR2 regulates protein *S*-nitrosylation in the mouse heart. (**A**) cGMP (pmol/mg total protein) in SCoR2$^{+/+}$ (+/+) and SCoR2$^{-/-}$ (-/-) mouse heart lysate post-I/R (1 hr reperfusion) as assessed by ELISA; *N* = 3 mouse hearts per group. (**B**) SNO-CoA (60 µM) added to +/+ mouse heart lysate (1 mg/ml) for 10 min increases protein *S*-nitrosylation as assessed by SNORAC and Coomassie blue staining, shown as fold increase relative to control after normalization to total protein; *N* = 10 +/+ mouse hearts per group. (**C**) In the presence of 100 µM NADPH (required for SCoR2 activity), cardiac protein *S*-nitrosylation is reduced, shown as fold decrease relative to SNO-CoA normalized to total protein; *N* = 10 +/+ mouse hearts per group. (**D**) Representative Coomassie-stained SDS/PAGE gel displaying SNO-proteins isolated by SNORAC following incubation of mouse heart extract with SNO-CoA alone or in combination with NADPH, as quantified in (**B**) and (**C**). Significance in (**A**) assessed by two-tailed Mann–Whitney test, and in (**B, C**) by ratio paired one-tailed Student's *t*-test. **p ≤ 0.01, ****p ≤ 0.0001.

The online version of this article includes the following source data and figure supplement(s) for figure 2:

**Source data 1.** Original western blots for *Figure 2D*, indicating the relevant bands and treatments.

**Source data 2.** Original files for western blot analysis displayed in *Figure 2D*.

**Figure supplement 1.** Characterization of SCoR2 and protein *S*-nitrosylation in SCoR2$^{-/-}$ mice.

**Figure supplement 1—source data 1.** JPEG file containing original western blots for *Figure 2—figure supplement 1B*, indicating the relevant bands and treatments.

**Figure supplement 1—source data 2.** Original files for western blot analysis displayed in *Figure 2—figure supplement 1B*.

**Figure supplement 1—source data 3.** Original western blots for *Figure 2—figure supplement 1D*, indicating the relevant bands and treatments.

**Figure supplement 1—source data 4.** Original files for western blot analysis displayed in *Figure 2—figure supplement 1D*.

**Figure supplement 1—source data 5.** Original western blots for *Figure 2—figure supplement 1E*, indicating the relevant bands and treatments.

**Figure supplement 1—source data 6.** Original files for western blot analysis displayed in *Figure 2—figure supplement 1EFigure 1—figure supplement 1*.

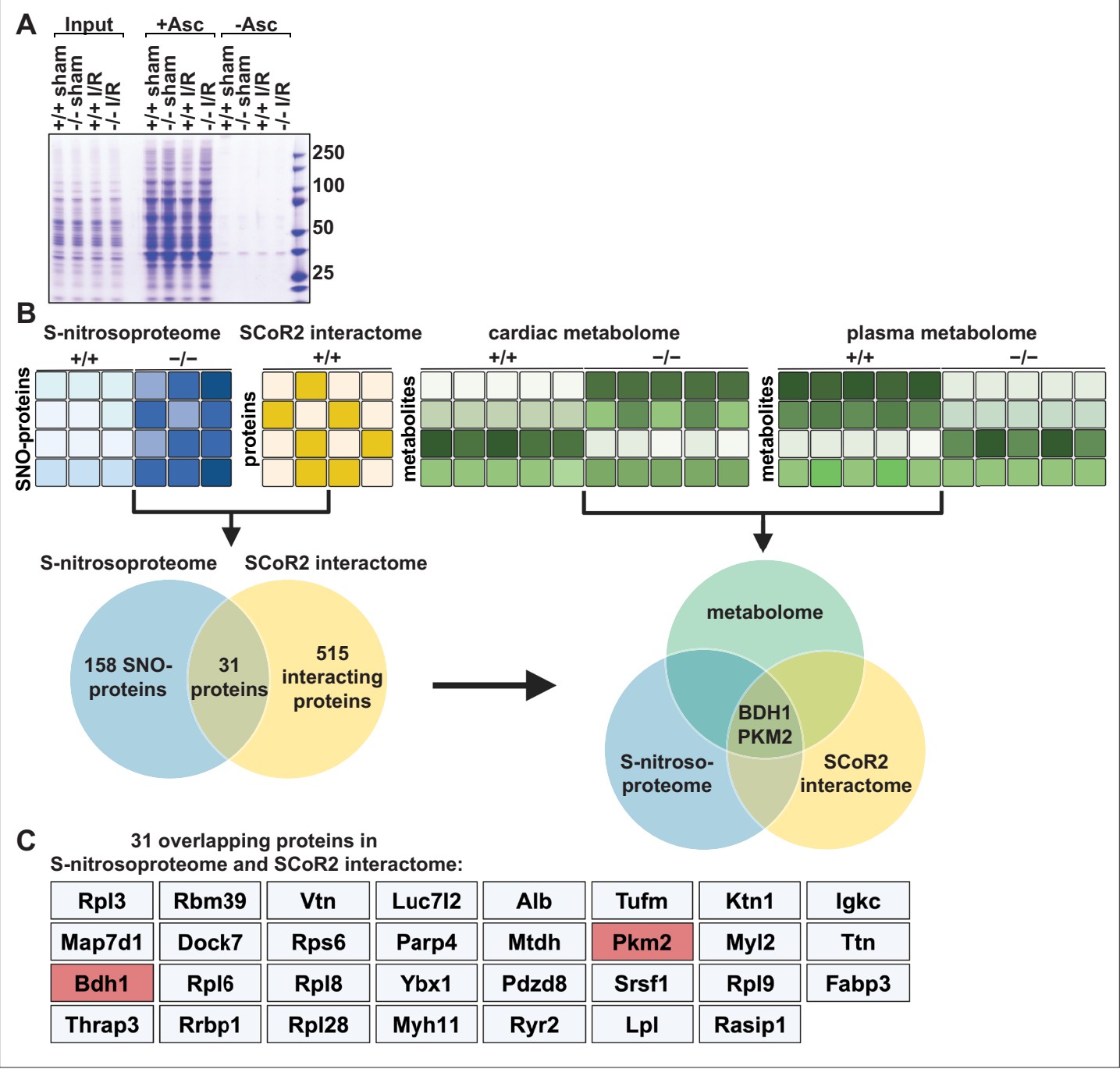

**Figure 3.** A combined multi-omics approach identifies SCoR2-regulated SNO-proteins and metabolic pathways responsible for widespread metabolic reprogramming. (**A**) *S*-Nitrosylated proteins in heart of SCoR2⁺/⁺ (+/+) and SCoR2⁻/⁻ (-/-) mice post-I/R versus sham. Representative Coomassie-stained SDS/PAGE gel displaying SNO-proteins isolated by SNORAC using hearts of +/+ and -/- mice subjected to either sham operation or I/R (4 hr reperfusion). Ascorbate was omitted from the SNORAC assay (-Asc) as a specificity control. (**B**) Three coordinated screens in +/+ versus -/- mouse heart tissue, that is (1) SNORAC/MS (SCoR2-dependent *S*-nitrosoproteome 4 hr after I/R, elevated >1.2-fold in -/- vs. +/+) (*Supplementary file 1*; N = 3), (2) SCoR2 co-IP interactome (*Supplementary file 1*; N = 4), and (3) untargeted metabolomic screening in heart and plasma (*Supplementary file 2*; N = 5 per condition), converge on the proteins BDH1 and PKM2 as SCoR2 substrate SNO-proteins in the heart which alter relevant cardioprotective metabolic pathways. (**C**) Full list of 31 overlapping proteins identified in both screens (1) and (2), that is, the cardiac SCoR2-dependent *S*-nitrosoproteome and the cardiac SCoR2 interactome. Panels B and C were created using Biorender.com.

The online version of this article includes the following source data for figure 3:

**Source data 1.** Original western blots for *Figure 3A*, indicating the relevant bands.

**Source data 2.** Original files for western blot analysis displayed in *Figure 3A*.

## SCoR2 regulates ketolytic energy availability through BDH1 stability

BDH1 catalyzes the first step of ketone body breakdown in the heart, the oxidation of β-hydroxy-butyrate (BHB) to acetoacetate, allowing cardiac muscle cells to use ketone bodies as an alternative source of energy in the absence of glucose and/or fatty acids (*Aubert et al., 2016*; *Nielsen et al., 2019*). Cardiac-specific BDH1 overexpression is known to be protective in mice subjected to cardiac injury, attributed to increased cardiac ketone body oxidation and associated increase in antioxidant defenses (*Uchihashi et al., 2017*).

S-Nitrosylation of BDH1 has been reported in proteomic screens (*Lau et al., 2021*) but remains entirely uncharacterized. In the post-I/R heart, we confirmed that SNO-BDH1 was increased in SCoR2$^{-/-}$ mice relative to SCoR2$^{+/+}$ (*Figure 4A*, quantified from Western blots in *Figure 4—figure supplement 1A*). Notably, total BDH1 protein expression was also increased in SCoR2$^{-/-}$ mice 4 hr after injury (*Figure 4B*, quantified from Western blots in *Figure 4—figure supplement 1A*), suggesting that SCoR2-controlled *S*-nitrosylation may regulate BDH1 stability. Indeed, the ratio of SNO-BDH1 to total BDH1 was not different in SCoR2$^{-/-}$ heart relative to SCoR2$^{+/+}$ (*Figure 4C*). Such regulation of SNO-BDH1 by SCoR2 appears specific to the heart, as levels of SNO-BDH1 and BDH1 were unchanged in post-I/R SCoR2$^{+/+}$ versus SCoR2$^{-/-}$ liver (*Figure 4D–F*, quantified from western blots in *Figure 4—figure supplement 1B*). To assess effects of *S*-nitrosylation on BDH1 stability, we first identified the predominant site of SNO modification as Cys$^{115}$ (out of seven total Cys residues), since in vitro SNO-CoA treatment failed to increase SNO-BDH1 in the Cys115Ser (C115S) mutant relative to WT BDH1 (*Figure 4G*). We next assessed BDH1 stability using a pulse-chase assay with the translation inhibitor cycloheximide (CHX chase) to measure the rate of BDH1 protein degradation in HEK293 cells. WT BDH1 protein expression decreased steadily over 12–24 hr (*Figure 4H*; black bars), while addition of a cell permeant NO donor (ethyl ester CysNO, ECNO) markedly increased BDH1 stability (pink bars). This stabilizing effect of NO was lost upon mutation of Cys$^{115}$ in BDH1 (*Figure 4H*; blue and purple bars, quantified as area under the curve in *Figure 4I*, with a representative western blot in *Figure 4—figure supplement 1C*), validating SNO-mediated stabilization of BDH1 and localizing the primary effect of NO to this single Cys residue.

We next sought the effects of SNO-BDH1 that might contribute to cardioprotection. We observed that acetoacetic acid, the ketone body produced by BDH1 in the heart, remained elevated in the plasma of SCoR2$^{-/-}$ mice after injury (*Figure 4J*), while acetoacetate level in the heart and BHB level in heart and plasma remained relatively unchanged between SCoR2$^{+/+}$ and SCoR2$^{-/-}$ (*Figure 4K*, *Figure 4—figure supplement 1D, E*). Metabolism of ketone bodies produces acetyl-CoA as an energetic intermediate, and we observed an increase in acetyl-CoA in hearts from SCoR2$^{-/-}$ mice (*Figure 4L*). Energy in the form of acetyl-CoA is used for ATP production, and excess ATP energy can be stored in muscle tissue as phosphocreatine (p-Cr). At the sarcoplasmic reticulum, p-Cr can be used rapidly to synthesize ATP in energy-starved states such as ischemia, supplying ATP for continued myocardial contraction, which is cardioprotective (*Landoni et al., 2016*). In the injured SCoR2$^{-/-}$ heart, we observed dramatically increased stored energy availability in the form of p-Cr, relative to SCoR2$^{+/+}$ at 4 hr post-I/R (*Figure 4M*). Together, these data support a role for increased energy availability in cardioprotection of SCoR2$^{-/-}$ mice after I/R injury (summarized in the model shown in *Figure 4N*).

## Regulated *S*-nitrosylation by SCoR2 in human hearts

BDH1 Cys$^{115}$ is conserved in mammals (*Figure 4—figure supplement 1F*), suggesting critical function. We therefore sought confirmation of a role for SNO-BDH1 in humans. *S*-nitrosylation of BDH1 was detected in human heart tissue, and SNO-BDH1 was elevated in hearts from patients with non-ischemic cardiomyopathy (NICM) (*Figure 4O*, representative image; quantified in *Figure 4P*). We reasoned that *S*-nitrosylation of BDH1 may be regulated by SCoR2 in humans. Expression of SCoR2 was, in fact, decreased in heart tissue from patients with NICM relative to healthy human heart tissue (*Figure 4Q, R*), suggesting an inverse relationship between SCoR2 activity and SNO-BDH1 levels, as seen in mice.

Interestingly, in patients with ischemic cardiomyopathy (ICM), SNO-BDH1 was decreased and SCoR2 levels were preserved, indicating context-specific regulation (and possibly maladaptive change; *Figure 4S, T*, representative SNORAC/western blot in *Figure 4—figure supplement 1*). Most importantly, we observed a significant correlation between SNO-BDH1 and total BDH1 (*Figure 4U*), suggesting *S*-nitrosylation may enhance BDH1 stability in human heart tissue, supporting our findings

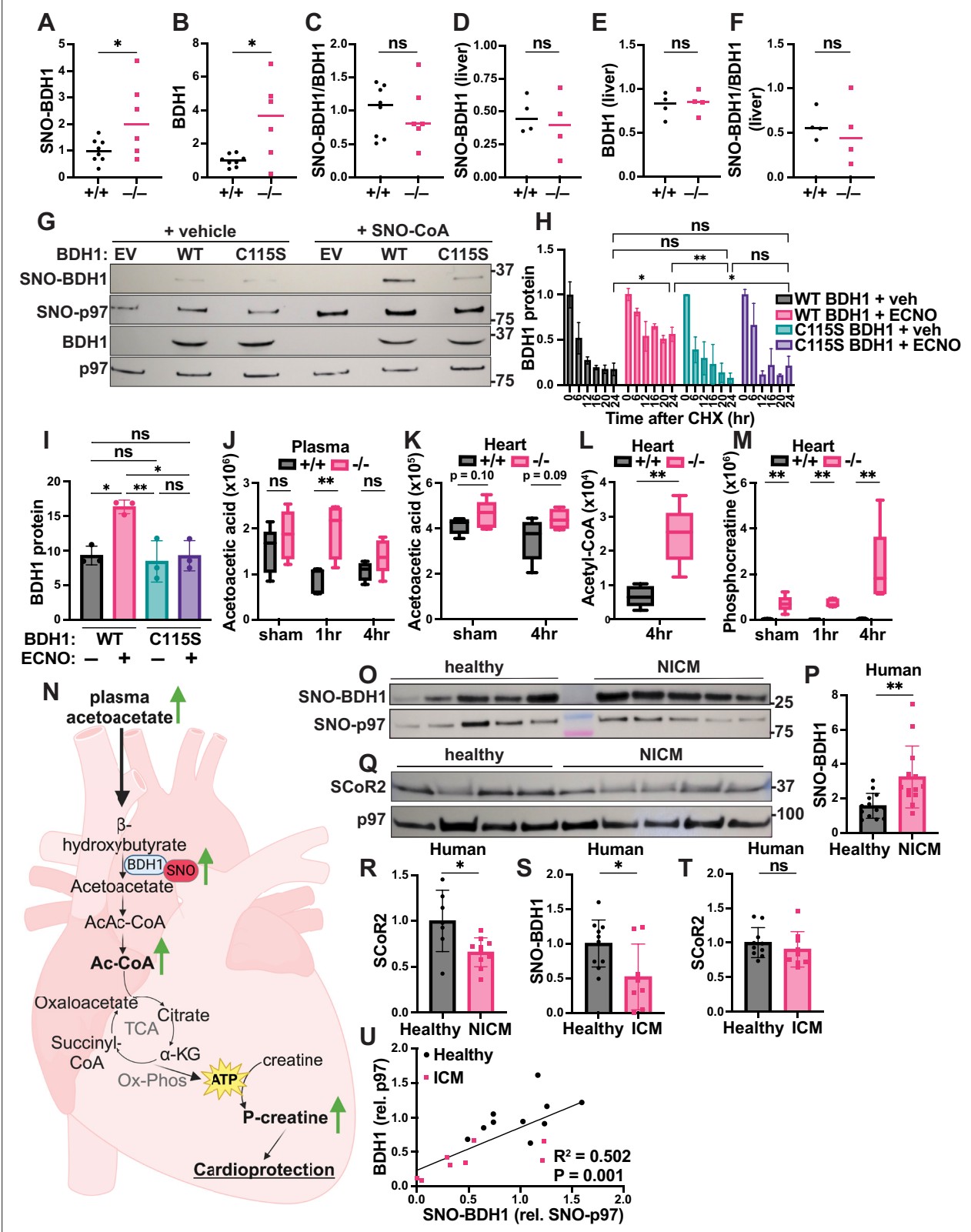

**Figure 4.** SCoR2 regulates *S*-nitrosylation of BDH1 at Cys[115], impacting ketolytic energy availability. Quantification of SNO-BDH1 (**A**) and total BDH1 (**B**), relative to p97 ATPase loading control and to each other (**C**), in mouse heart (4 hr reperfusion); $N$ = 8 SCoR2[+/+] (+/+), $N$ = 6 SCoR2[-/-] (-/-). Representative western blots shown in *Figure 4—figure supplement 1A*. Quantification of SNO-BDH1 (**D**) and total BDH1 (**E**), relative to p97 ATPase loading control and to each other (**F**), in mouse liver (1 hr reperfusion); $N$ = 4 each. (**G**) HEK293 cells transfected with empty vector (EV), WT, or C115S

*Figure 4 continued on next page*

*Figure 4 continued*

BDH1 were treated with 250 μM SNO-CoA or vehicle for 10 min, followed by SNORAC and blotted for SNO-BDH1 and BDH1, together with loading controls SNO-p97 ATPase and input p97 ATPase. Representative of four independent experiments. (**H, I**) CHX pulse-chase assay, in which HEK293 cells transfected with V5-tagged BDH1 were treated with 100 μM ECNO or vehicle, in addition to 100 μg/ml CHX for 6–24 hr. BDH1 protein expression was quantified relative to t=0 (pre-CHX) in (**H**), and area under the curve (AUC) quantified in (**I**); *N* = 3 samples/group. (**J–M**) Metabolites corresponding to ketolytic energy availability in mouse heart and plasma subjected to sham or I/R injury (1 or 4 hr reperfusion), quantified as ion abundance by LC/MS-based untargeted metabolite profiling: (**J**) plasma acetoacetic acid, (**K**) heart acetoacetic acid, (**L**) heart acetyl-CoA, (**M**) heart phosphocreatine. *N* = 5 mice per condition per genotype. (**N**) Model depicting effect of SCoR2 deletion on cardiac ketone body metabolism in SCoR2$^{-/-}$ mice. (**O–R**) Human heart samples (IRB# Pro00005621, *N* = 13 with diagnosis of non-ischemic cardiomyopathy (NICM) and *N* = 13 without known cardiac pathophysiology (healthy)) subjected to SNORAC measuring SNO-BDH1 expression relative to loading control (SNO-p97 ATPase), representative SNORAC shown in (**O**), quantification in (**P**). (**Q, R**) SCoR2 protein expression in human heart samples (*N* = 7 healthy, *N* = 10 NICM from same cohort), as determined by western blot relative to p97 ATPase (representative image in (**Q**), quantified in (**R**)). (**S–U**) Human heart samples (IRB# Pro00005621, *N* = 8–9 with diagnosis of ischemic cardiomyopathy (ICM; pink) and *N* = 10 without known cardiac pathophysiology (healthy; black)) subjected to SNORAC measuring SNO-BDH1 (**S**) or western blot measuring SCoR2 relative to loading control (SNO-p97 ATPase or p97, respectively). Representative SNORAC/western blot gels shown in *Figure 4—figure supplement 1G, H*. Correlation of SNO-BDH1 (normalized to SNO-p97) versus BDH1 (normalized to p97) expression in healthy (*N* = 10) and ICM (*N* = 8) heart was assessed by simple linear regression in (**U**). Statistical significance in (**A–C**) determined by two-tailed Student's *t*-test; (**D–F**) determined by two-tailed Mann–Whitney test; (**H**) determined by two-way ANOVA with Tukey's multiple comparisons test; (**I**) determined by one-way ANOVA with Tukey's multiple comparisons test; (**J–L**) determined by multiple independent Student's *t*-tests performed between genotypes in each condition; (**M, P**) determined by two-tailed Mann–Whitney test; (**R–T**) determined by Student's *t*-test; (**U**) determined by simple linear regression performed to identify SNO-BDH1 versus BDH1 relationship. $R^2$ and p-value show the goodness of fit of the regression model and significance of slope difference from zero, respectively. *p ≤ 0.05, **p ≤ 0.01, ns = not significant.

The online version of this article includes the following source data and figure supplement(s) for figure 4:

**Source data 1.** Original western blots for *Figure 4G*, indicating the relevant bands.

**Source data 2.** Original files for western blot analysis displayed in *Figure 4G*.

**Source data 3.** Original western blots for *Figure 4O*, indicating the relevant bands.

**Source data 4.** Original files for western blot analysis displayed in *Figure 4O*.

**Source data 5.** Original western blots for *Figure 4Q*, indicating the relevant bands.

**Source data 6.** Original files for western blot analysis displayed in *Figure 4Q*.

**Figure supplement 1.** Characterization of BDH1 *S*-nitrosylation at conserved SNO site Cys$^{115}$.

**Figure supplement 1—source data 1.** Original western blots for *Figure 4—figure supplement 1* (panel A), indicating the relevant bands.

**Figure supplement 1—source data 2.** Original files for western blot analysis displayed in *Figure 4—figure supplement 1* (panel A).

**Figure supplement 1—source data 3.** Original western blots for *Figure 4—figure supplement 1* (panel B), indicating the relevant bands.

**Figure supplement 1—source data 4.** Original files for western blot analysis displayed in *Figure 4—figure supplement 1* (panel B).

**Figure supplement 1—source data 5.** Original western blots for *Figure 4—figure supplement 1* (panel C), indicating the relevant bands.

**Figure supplement 1—source data 6.** Original files for western blot analysis displayed in *Figure 4—figure supplement 1* (panel C).

**Figure supplement 1—source data 7.** Original western blots for *Figure 4—figure supplement 1* (panel G), indicating the relevant bands.

**Figure supplement 1—source data 8.** Original files for western blot analysis displayed in *Figure 4—figure supplement 1* (panel G).

**Figure supplement 1—source data 9.** Original western blots for *Figure 4—figure supplement 1* (panel H), indicating the relevant bands.

**Figure supplement 1—source data 10.** Original files for western blot analysis displayed in *Figure 4—figure supplement 1* (panel H).

in cell culture (*Figure 4H, I*). Altogether, our data demonstrate that *S*-nitrosylation of metabolic proteins may correlate with human disease, suggests that SNO-BDH1 may be a protective locus that is regulated by SCoR2, and identifies a possible signature of SCoR2 deficiency in human NICM.

## SCoR2 regulates glycolytic and PPP intermediates through *S*-nitrosylation of PKM2

PKM2 is the injury-upregulated splice variant of muscle-type pyruvate kinase, which includes a regulatory SNO-site controlling PKM2 conversion of phosphoenolpyruvate to pyruvate in glycolysis. *S*-nitrosylation of PKM2 at Cys$^{423/424}$ inhibits glycolytic flux, thereby shunting glycolytic intermediates into the PPP (*Zhou et al., 2019*; *Siragusa et al., 2019*) to produce antioxidant NADPH along with nucleotide precursors. The PPP can also generate newly identified polyol precursors subject to additional SCoR2 regulation, as detailed below. PKM2 is upregulated by ischemic injury, and SNO-PKM2 is known to be protective in the context of ischemia in the kidney (*Zhou et al., 2019*) and of oxidative

stress in the endothelium (*Siragusa et al., 2019*). Notably, hearts of patients with ICM showed a significant increase in SNO-PKM2 (*Figure 5A*; representative SNORAC and western blot in *Figure 4—figure supplement 1G, H*), consistent with a cardioprotective effect.

In 4 hr post-I/R heart, the level of lactate, a product of inefficient glycolysis during ischemia that is associated with worse clinical outcomes in patients with MI (*Lazzeri et al., 2015*), was markedly reduced (*Figure 5B*), suggesting reduced activity of the glycolysis pathway. More directly, we observed increased glycolytic and non-glycolytic input into the PPP in SCoR2$^{-/-}$ heart (*Table 1*), including plasma pyruvate (*Figure 5C*) and arabinose (*Figure 5D*). Pyruvate is produced through multiple metabolic pathways, including non-glycolytic pathways, and can enter the PPP through conversion to xylose. Together, these findings indicate that metabolites available to the PPP are increased in the absence of SCoR2, consistent with inhibition of PKM2 by *S*-nitrosylation (*Figure 3B*) (as observed in multiple tissues [*Zhou et al., 2019*; *Siragusa et al., 2019*] and known to be protective in the setting of ischemia [*Zhou et al., 2019*]). We also observed that products of the PPP were increased in SCoR2$^{-/-}$ heart, including NADPH (*Figure 5E*) and erythrose 4-phosphate (*Figure 5F*), as well as increased ribose (*Figure 5G*) and deoxyribose (*Figure 5H*) in SCoR2$^{-/-}$ plasma. Altogether, this metabolic signature is consistent with the previously described inhibition of PKM2 by *S*-nitrosylation in the absence of SCoR2 (*Zhou et al., 2019*).

## SCoR2 regulates endogenous production of harmful polyol compounds

Regulation of polyol metabolism by SCoR2 may contribute to protection from myocardial ischemia. Elevated levels of polyols, organic metabolites with >1 hydroxyl group, have recently been associated with short-term (<3 year) risk for major adverse cardiovascular events (*Witkowski et al., 2023*; *Witkowski et al., 2024*). We observed that two polyols, arabitol/ribitol and sorbitol, were markedly decreased in plasma from SCoR2$^{-/-}$ mice relative to SCoR2$^{+/+}$ (*Figure 5I, J*). Additionally, plasma levels of each polyol rose at 4 hr post-I/R relative to sham or 1 hr post-I/R in SCoR2$^{+/+}$ mice, but not in SCoR2$^{-/-}$ mice.

Endogenous polyol production remains relatively uncharacterized in humans. Some polyols may derive from PPP metabolites (xylulose, ribulose, ribose, erythrose, etc.) (*Wamelink and Williams, 2022*; *Huck et al., 2004*) via actions of reductase enzymes (*Hamada et al., 1991*), but rigorous evidence is lacking. To test whether SCoR2 might directly generate polyols that we find to be elevated in the absence of SCoR2 (arabitol/ribitol, sorbitol; *Figure 5I, J*), we measured reduction of the corresponding carbohydrates (i.e., arabinose, ribose, glucose, and fructose) in the presence of recombinant SCoR2 (vs. the canonical SCoR2 substrate SNO-CoA as positive control). Using multiple carbohydrates known or predicted to form polyols upon reduction as potential substrates, SCoR2 demonstrated no direct reductase activity as measured by NADPH consumption (*Figure 5K*). The related enzyme AKR1B1 can reduce glucose to the polyol sorbitol, but glucose is not a good substrate for AKR1A1/SCoR2 *Fujii et al., 2021*; *Danesh et al., 2003* and was not reduced in our assay (*Figure 5K*).

## Alternative substrates of SCoR2 in polyol metabolism

SCoR2 has been formally classified as an aldoketoreductase. However, putative carbonyl substrates of SCoR2/AKR1A1 have not been validated in situ and were not detectable in SCoR2$^{-/-}$ heart or plasma (*Supplementary file 2*). These data are consistent with SCoR2 functioning primarily as a denitrosylase to regulate polyol metabolism via multiple substrate enzymes, rather than as a direct carbohydrate reductase. Taken together, our findings lead to an overall model (*Figure 5L*) in which SCoR2 reprograms cardiac metabolism through multiple metabolic pathways including cardiac ketolysis, glycolysis, PPP, and polyol metabolism, to increase protective NADPH and phosphocreatine in the mouse heart, while decreasing lactate and multiple polyols that are associated with poor cardiovascular outcomes.

## Discussion

The cardioprotective role of NO in pre-clinical models of acute MI is well established (*Gonzalez et al., 2008*; *Dezfulian et al., 2009*; *Duranski et al., 2005*; *Hendgen-Cotta et al., 2008*). However, the ability of NO donors to reduce I/R damage has been disappointing in the clinical setting (*Siddiqi et al., 2014*; *Jones et al., 2015*). One explanation for these failures is the absence of drug targets or cardioprotective mechanisms by which to assess efficacy of NO donors. This challenge is circumvented

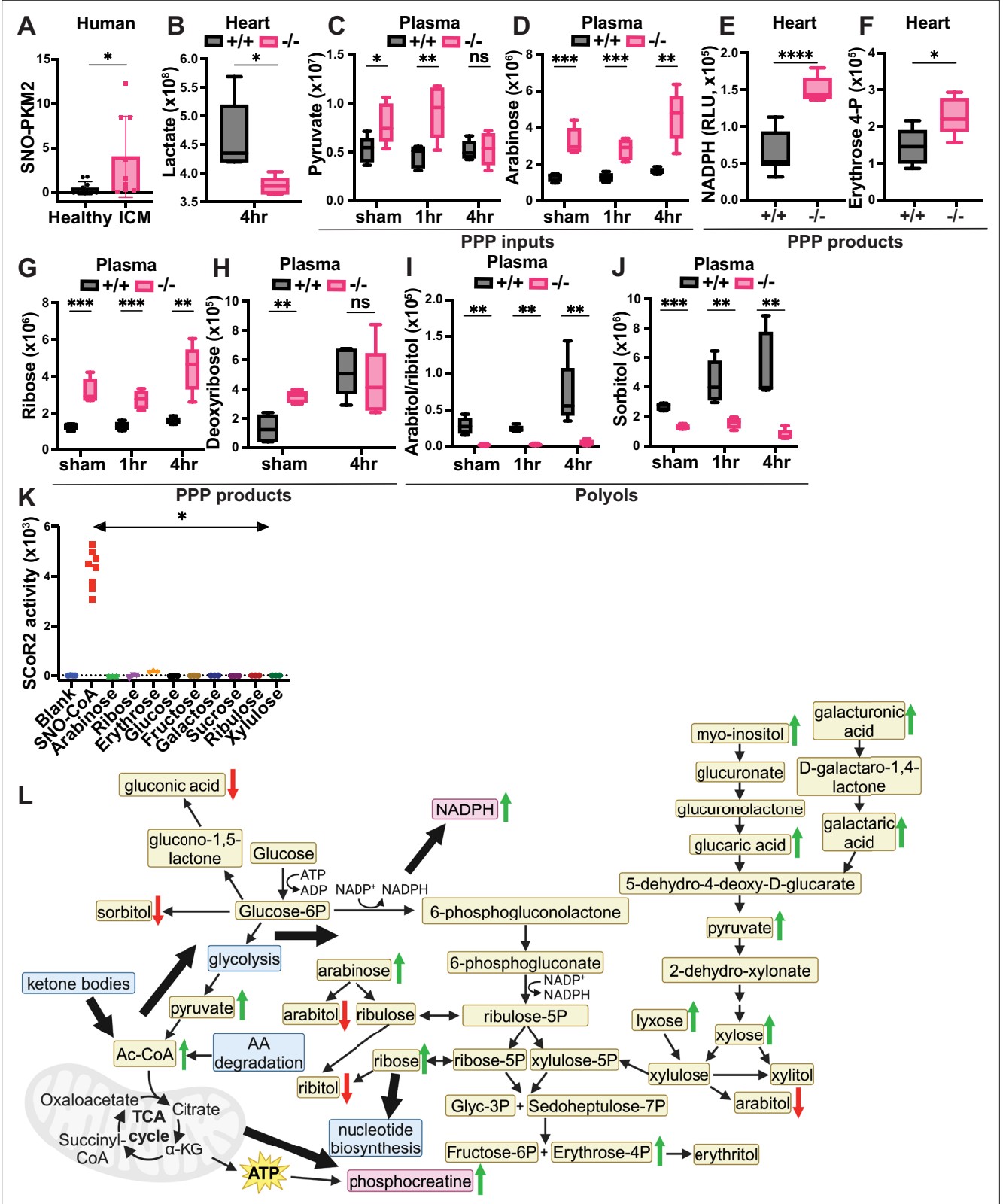

**Figure 5.** SCoR2 regulates carbohydrate metabolism, including polyols, to provide antioxidative protection from injury via the pentose phosphate shunt pathway (PPP). (**A**) Human heart samples (IRB# Pro00005621, *N* = 9 with diagnosis of ischemic cardiomyopathy (ICM) and *N* = 10 without known cardiac pathophysiology (healthy)) subjected to SNORAC measuring SNO-PKM2 relative to loading control (SNO-p97 ATPase). Representative SNORAC/Western blot gels shown in *Figure 4—figure supplement 1G, H*. (**B–J**) Metabolites quantified by ion abundance in SCoR2^+/+ (+/+) and

*Figure 5 continued on next page*

*Figure 5 continued*

SCoR2$^{-/-}$ (-/-) mouse heart and plasma by LC/MS-based untargeted metabolite profiling; *N* = 5 each condition. (**B**) Heart lactate. (**C–K**) Metabolites organized by relationship to the PPP, as inputs (**C, D**), products (**E–H**), or polyol compounds that are categorized as downstream end products of the PPP (**I, J**). NADPH measured in heart lysate after 2 hr reperfusion, erythrose 4-phosphate measured after 4 hr reperfusion. (**K**) Recombinant SCoR2 activity quantified via NADPH consumption by spectrophotometer in the presence of canonical substrate (100 μM SNO-CoA) or carbohydrates (1 mM); [SCoR2]=186 nM, [NADPH]=100 μM. Results presented as specific activity of SCoR2 (μM substrate consumed/min/mg protein). Assay performed in triplicate. (**L**) Summary model showing SCoR2-mediated regulation of carbohydrate metabolism, including PPP and polyol compounds, to generate NADPH and phosphocreatine in the mouse heart. Green arrows indicate pathway upregulation in the absence of SCoR2, and red arrows indicate downregulation. Blue boxes indicate metabolic pathways, tan boxes indicate metabolites, and pink boxes indicate metabolites of particular significance. Thick black arrows indicate directions of SCoR2-regulated metabolic changes. This panel was created using Biorender.com. Statistical significance in (**A**) determined by Student's *t*-test; (**B–H, J**) determined by multiple independent Student's *t*-tests performed between genotypes in each condition; (**I**) determined by two-tailed Mann–Whitney test; (**K**) determined by two-tailed Mann–Whitney test between SNO-CoA condition and each carbohydrate condition (*N* = 3–8 independent replicates per condition). *p ≤ 0.05, **p ≤ 0.01, ***p ≤ 0.001, ****p ≤ 0.0001.

through enzymatically regulated *S*-nitrosylation, where the target enzyme has been genetically validated. In particular, the denitrosylase SCoR2 (*Anand et al., 2014*) regulates multiple metabolic pathways, including ketolysis, glycolysis, PPP, and polyol metabolism, that have established roles in cardioprotection, but were not previously known to be coordinately regulated. Our studies thus identify a comprehensive role for SCoR2 in regulating cardiac metabolism and suggest a new approach to cardioprotection through global metabolic reprogramming.

A large body of evidence supports the benefits of ketones in acute myocardial injury. BDH1 catalyzes the initial step in the breakdown of the ketone BHB in extrahepatic tissues, and cardiac-specific knockout of BDH1 in mice leads to more severe ventricular dysfunction following ischemia (*Horton et al., 2019*). Conversely, cardiac-specific BDH1 overexpression improves contractile function in response to injury (*Uchihashi et al., 2017*). Ketone bodies secreted from the liver are a critical source of myocardial energy during fasting (*Mitchell et al., 1995*; *Homilius et al., 2023*; *Cotter et al., 2013*) and also in the failing (*Aubert et al., 2016*; *Nielsen et al., 2019*) and ischemic heart (*Zuurbier et al., 2020*), where they reduce infarct size and improve post-I/R contractile function (*Marina Prendes et al., 2005*; *Snorek et al., 2012*; *Hron et al., 1978*). Exogenous BHB reduces infarct size in rats (*Zou et al., 2002*) and mice (*Yu et al., 2018*) subjected to I/R injury by increasing ATP production and is beneficial in patients with cardiogenic shock (*Berg-Hansen et al., 2023*), providing evidence that ketolysis contributes to superior metabolism in the setting of acute I/R injury. Circulating ketone

**Table 1.** Input metabolites to the pentose phosphate pathway (PPP) via xylulose.
Relative quantity, quantified in mean ion abundance ± standard deviation by LC/MS-based untargeted metabolomic platform, as described in the Methods.

| Metabolite | Organ | SCoR2$^{+/+}$ | | | SCoR2$^{-/-}$ | | |
|---|---|---|---|---|---|---|---|
| | | sham | MI (1 hr) | MI (4 hr) | sham | MI (1 hr) | MI (4 hr) |
| Galactaric acid | Heart | 4.88E+04 +/- 2.14E+04 | 7.08E+04 +/- 3.62E+04 | 2.16E+04 +/- 1.64E+04 | 1.09E+05 +/- 2.37E+04 | 1.17E+05 +/- 2.44E+04 | 2.84E+05 +/- 2.13E+05 |
| Galactaric acid | Plasma | 4.90E+05 +/- 1.83E+05 | 3.83E+05 +/- 8.86E+04 | 8.05E+05 +/- 2.16E+05 | 6.14E+06 +/- 1.84E+06 | 5.82E+06 +/- 2.11E+06 | 1.42E+07 +/- 6.43E+06 |
| Galacturonic acid | Heart | 4.70E+04 +/- 4.03E+04 | 5.53E+04 +/- 2.38E+04 | 1.24E+05 +/- 2.36E+04 | 3.71E+05 +/- 1.63E+05 | 3.47E+05 +/- 8.03E+04 | 9.85E+05 +/- 5.72E+05 |
| Galacturonic acid | Plasma | 6.06E+07 +/- 2.18E+07 | 5.32E+07 +/- 8.19E+06 | 1.54E+08 +/- 3.76E+07 | 5.05E+08 +/- 9.52E+07 | 4.42E+08 +/- 7.53E+07 | 8.55E+08 +/- 2.53E+08 |
| Glucaric acid | Heart | 3.08E+05 +/- 1.36E+05 | 3.31E+05 +/- 9.75E+04 | 4.46E+05 +/- 1.74E+05 | 9.58E+06 +/- 4.84E+06 | 9.93E+06 +/- 2.53E+06 | 3.35E+07 +/- 2.68E+07 |
| Glucaric acid | Plasma | 2.13E+06 +/- 6.93E+05 | 2.26E+06 +/- 7.27E+05 | 5.92E+06 +/- 1.28E+06 | 1.81E+08 +/- 3.10E+07 | 1.66E+08 +/- 4.77E+07 | 2.68E+08 +/- 9.96E+07 |
| Xylose | Plasma | 1.36E+06 +/- 1.95E+05 | 1.35E+06 +/- 2.51E+05 | 1.68E+06 +/- 1.61E+05 | 3.37E+06 +/- 7.05E+05 | 2.91E+06 +/- 5.66E+05 | 4.70E+06 +/- 1.35E+06 |
| Lyxose | Plasma | 1.08E+06 +/- 1.69E+05 | 1.10E+06 +/- 1.88E+05 | 1.39E+06 +/- 1.71E+05 | 2.84E+06 +/- 6.11E+05 | 2.41E+06 +/- 4.26E+05 | 4.02E+06 +/- 1.20E+06 |

bodies are elevated in humans with congestive heart failure (*Lommi et al., 1996*) and are associated with improved cardiac function (*Selvaraj et al., 2020*) and a lower risk of adverse events in acute MI (*Sato et al., 2023*). Cardiac ketone body metabolism, beginning with BDH1-mediated breakdown of BHB, is thus a validated protective pathway in acute I/R injury and is beneficial in the context of human cardiovascular disease.

In this context, we have discovered that SCoR2 regulates *S*-nitrosylation of cardiac BDH1 at Cys[115], thereby governing its stability. BDH1 expression is thus increased in SCoR2[-/-] hearts, in accord with its increased *S*-nitrosylation. Increased expression of BDH1 increases capacity for ketolysis, improving energy reserves (i.e., p-Cr) during ischemia (*Homilius et al., 2023*), as we confirm here. Many studies, including over 40 controlled trials in patients with CAD, HF, or cardiac surgery (*Landoni et al., 2016*), have demonstrated that p-Cr is cardioprotective (*Landoni et al., 2016*; *Sharov et al., 1986*; *Prabhakar et al., 2003*; *Zhang et al., 2015*; *Qaed et al., 2019*; *Wang et al., 2021*), an effect attributed to increased energy availability. Strategies to increase p-Cr in the post-ischemic myocardium are thus a promising avenue to reduce post-MI tissue damage. In this light, we show that inhibition of SCoR2 results in increased p-Cr.

PKM2 is a pyruvate kinase splice variant expressed in development, injury, and cancer (*Cheon et al., 2016*; *Wang et al., 2012*). PKM2 inhibition is strongly linked to tissue protection, regeneration, and repair (*Zhou et al., 2019*; *Siragusa et al., 2019*; *Hauck et al., 2021*; *Cheng et al., 2017*), including in the heart where inhibition (*Cheng et al., 2017*) or genetic deletion (*Hauck et al., 2021*) leads to reduced infarct size. *S*-nitrosylation of PKM2 inhibits activity and thus progression of intermediates through glycolysis (*Zhou et al., 2019*); instead, glucose 6-phosphate is shunted into the PPP, where it is used to synthesize the antioxidant NADPH and 5-carbon sugars for biosynthesis (*TeSlaa et al., 2023*). This is protective in the setting of MI (*Hauck et al., 2021*; *Cheng et al., 2017*). The PPP is responsible for producing a portion of the cardiac NADPH pool, and this pathway is upregulated in stressed or injured cardiac tissue (*Gupte et al., 2006*; *Wang et al., 2022*; *Zoccarato et al., 2023*). SNO-PKM2, a validated SCoR2 substrate, protects against ischemic kidney injury and endothelial damage (*Siragusa et al., 2019*) via increased flux through the PPP shunt (*Zhou et al., 2019*; *Zhou et al., 2023b*). We demonstrate that deletion of SCoR2 leads to an increase in SNO-PKM2 in the heart, accompanied by inhibition of glycolysis (decreased lactate) and stimulation of the PPP (increased NADPH, erythrose 4-phosphate, and ribose). We also observe an increase in other inputs into the PPP (primarily those converging on xylulose, such as galactaric acid, xylose, and lyxose, among others), supporting a comprehensive role for SCoR2 in regulation of the protective PPP shunt in the context of acute ischemic injury.

Polyols are relatively understudied organic metabolites containing more than 1 hydroxyl group. The 'polyol pathway' historically refers specifically to sorbitol *Garg and Gupta, 2022*; however, there are many other polyols that are metabolically relevant in mammals, as first described in the mid-20th century (*Hutcheson et al., 1956*; *Touster and Harwell, 1958*). Endogenous production of polyols, well characterized in microorganisms (*Lactobacillus*, *E. avium*, *L. casei*, and others) (*Rice et al., 2020*; *Monedero et al., 2010*), has remained underappreciated and relatively uncharacterized in humans. Recent evidence of endogenous erythritol synthesis in human blood (*Hootman et al., 2017*), and in A549 lung cancer cells by the reductases ADH and SORD (*Schlicker et al., 2019*), casts doubt on the view that polyols are not endogenously produced in humans (*Hiele et al., 1993*). Additionally, heritable genetic mutations (i.e., in transketolase *Boyle et al., 2016*, transaldolase *Verhoeven et al., 2001*, and ribose-5-phosphate isomerase *Huck et al., 2004*) may lead to aberrant accumulation of certain polyols (ribitol, arabitol, and erythritol), suggesting close connections to the PPP. Many polyols are structurally related to PPP intermediates, such as arabitol, xylitol, and ribitol, and are likely derived at least in part from these compounds (*Wamelink and Williams, 2022*; *Huck et al., 2004*). Most studies appear to presume that any polyols produced are simply excreted from the body (*Boyle et al., 2016*).

However, there is tantalizing new evidence that polyols impact cardiovascular health (*Witkowski et al., 2023*; *Witkowski et al., 2024*). Erythritol, xylitol, threitol, arabitol, and myo-inositol are highly associated with incident risk for major adverse cardiovascular events, and erythritol, in particular, is reported to enhance in vitro platelet reactivity and in vivo thrombosis rate (*Witkowski et al., 2023*). Our untargeted metabolomic analysis revealed that SCoR2 deletion causes a marked reduction in these polyol 'off-shoots' from the PPP, particularly arabitol/ribitol (indistinguishable in our analysis)

and sorbitol, a difference that widens after 4 hr of reperfusion relative to sham injury. We present strong evidence that this effect is not mediated through direct reduction of carbohydrates by SCoR2, in contrast with a recent report (*Hoshino et al., 2024*) (where the actual data are negative, supporting our findings) and literature viewpoint. Rather, the many observed changes in carbohydrates and corresponding polyols are seemingly well rationalized by the widespread alterations in *S*-nitrosylation of metabolic enzymes (*Supplementary file 1*) accompanying SCoR2 deletion. Indeed, multiple reductase enzymes were identified in the SCoR2-dependent nitrosoproteome (e.g., ALDH1B1 and DHRS7B, *Supplementary file 1*), some of which may catalyze direct reduction of carbohydrates to polyols in an *S*-nitrosylation- and SCoR2-dependent manner. Multiple reductase and aldolase enzymes were recently identified as interacting partners with polyols (such as arabitol) and related carbohydrates (such as arabinose, ribulose, xylose, ribose, and xylulose) (*Hicks et al., 2023*), including aldolase B, MDH2, PDH, and IDH3 that all have been identified in SNO-proteomic screens (*Mnatsakanyan et al., 2019*) and may be SCoR2-regulated. Reduction in polyol levels thus represents a fourth SCoR2-regulated metabolic pathway contributing to comprehensive metabolic reprogramming during I/R injury.

The emerging concept that protein *S*-nitrosylation can be regulated independently of NO synthesis (i.e., by SNO metabolism) changes the therapeutic paradigm in myocardial ischemia. Protein denitrosylases, including both GSNOR (*Lima et al., 2009*; *Tang et al., 2023*; *Castillo et al., 2021*; *Grimmett et al., 2021*; *Hatzistergos et al., 2015*) and thioredoxin (*Perone and Lembo, 2020*; *Tao et al., 2004*), have been shown to play important roles in cardiac injury and repair. However, a general schema for how these enzymes confer protection has been missing. Here we establish a cardioprotective framework for SCoR2 through metabolic regulation, whereby SCoR2 inhibition coordinately orchestrates multiple metabolic pathways for therapeutic gain: stimulation of ketolysis, inhibition of glycolysis, increased PPP shunting, and reduction in polyol compounds converge to protect the heart (*Figure 5L*). These findings help to cement SCoR2 as a major regulator of metabolism in the injured myocardium and new class of pharmaceutical target.

## Limitations

The results presented here are purposefully limited to the acute response to cardiac tissue injury and do not explore whether deletion or inhibition of SCoR2 may be beneficial in long-term recovery from MI. Metabolic regulation in chronic injury may involve alternative pathways. The protection offered by SCoR2-mediated metabolic reprogramming is also specific to MI injury, although suggestive findings are shown in human cardiomyopathy. We identified the critical enzymes PKM2 and BDH1 via unbiased multi-omic screens, and we surmise they coordinately regulate increased flux through the PPP and ketolysis pathways upon function-regulating *S*-nitrosylation, thereby explaining improved energy balance, but isotope tracing experiments would be necessary to confirm this.

## Materials and methods

### Cell lines

HEK293 cells (ATCC CRL-1573) were obtained from ATCC and cultured in Dulbecco's modified Eagle's medium (DMEM) containing 10% fetal bovine serum and antibiotic/antimycotic solution (Gibco 15240096) (complete DMEM). All cells were grown at 37°C under 5% $CO_2$.

### Plasmid construction and mutagenesis

The BDH1 (human) cDNA in pDONR223 entry vector was obtained from DNAsu (clone:HsCD00352477). This entry vector was cloned into Gateway pcDNA-DEST40 (mammalian expression, under CMV promoter; cat#43-0101; Invitrogen) with V5 epitope tag, using Gateway LR Clonase II (cat#11791-020; Invitrogen) enzyme reaction, and the LR reaction was transformed into One Shot Omnima cells (cat# C8540-03; Invitrogen) and streaked onto selective Amp+ plates. A single colony was isolated, expanded in LB + Amp media, and plasmid purified by QIAGEN Plasmid Maxi Kit (cat#12362; QIAGEN). The resulting plasmid was sequenced to confirm correct gene and vector sequence. C115S-BDH1 mutant was generated using QuikChange II Site-Directed Mutagenesis Kit (cat# 200523; Agilent) using mutagenesis primers (forward: 5'-CGTCCAGCTCAATGTCTCCAGCAGCGAAGAGG; reverse: 5'-CCTC

TTCGCTGCTGGAGACATTGAGCTGGACG) from Invitrogen. Empty vector (pcDNA3.1(+); Invitrogen) was expanded and sequenced as described above.

## Transfection

For BDH1 transfections, HEK293 cells were plated onto dishes coated with 5 mg/cm$^2$ poly-D-lysine (Corning) and cultured for 24 hr in growth media. Cells were transfected with indicated empty vector, WT, or C115S BDH1 plasmids using PolyJet transfection reagent (Signagen) per manufacturer's instructions. After 24 hr, cells were treated with vehicle or 100 µM Ethyl ester SNO-cysteine (ECNO), which was freshly prepared by mixing equal volumes of 2 M sodium nitrite with 2 M acidified ethyl ester cysteine (0.5 M HCl) (Millipore-Sigma) to generate 1 M ECNO, which was immediately added to designated cells at a final concentration of 100 µM. The vehicle was prepared in the same manner, with omission of ethyl ester cysteine from 0.5 M HCl. At the completion of the experiment, cells were harvested, pelleted, washed 1x with cold PBS, and subsequently resuspended in NP40 lysis buffer for biochemical analysis.

## Cycloheximide pulse-chase assay

Cycloheximide pulse-chase assay (CHX chase) was performed using CHX (stock concentration: 10 mg/ml in ddH$_2$O; Sigma-Aldrich, cat#01810) added directly to cell culture media (at the time of ECNO/vehicle addition) to a final concentration of 100 µg/ml for the indicated time period. At the end of the indicated time period, cells were harvested, washed 1x with cold PBS, and immediately frozen at –80°C. Once all time points were collected, all samples were thawed and immediately lysed in NP40 lysis buffer for biochemical analysis.

## Cloning, expression, and purification of recombinant SCoR2

The human *SCoR2/AKR1A1* coding sequence was previously cloned into the pET21b vector (*Anand et al., 2014*) to introduce a C-terminal 6xHis tag on the expressed protein. The recombinant SCoR2 protein was purified from BL21-CodonPlus Competent *E. coli* cells (Agilent). Overnight *E. coli* cultures were sub-cultured into 1 l of LB medium at 5%. At OD600 of 0.5, cultures were induced with 1 mM IPTG (Invitrogen) and grown for a further 4 hr at 28°C. Cultures were centrifuged at 4000 × *g* for 10 min to harvest the cells. Cell pellets from 1 l cultures were lysed in 10 ml of 1x PBS buffer containing 1 mM PMSF (cat# P7626; Sigma), protease-inhibitor cocktail (Roche), DNAse 1 (cat# 260913; Sigma-Aldrich; final: 5 µg/ml) and lysozyme (Sigma, cat#L6876; final: 1 mg/ml) by sonication. Supernatant was clarified and SCoR2-His purified via HisTrap column (Cytiva; cat#17524801); binding with 20 mM imidazole and elution with 500 mM imidazole using Cytiva AKTA Pure chromatography system, followed by buffer exchange to assay buffer (50 mM borate, 0.1 mM EDTA, pH 9.0). Protein was concentrated using Ultra-15 centrifugal filters (Amicon), and concentration was determined by Bicinchoninic Acid (BCA) assay (Thermo Scientific; cat#23223/4). Purification was assessed by SDS-PAGE followed by Imperial Protein Stain (Thermo Fisher).

## Mice

Mouse studies were approved by the Case Western Reserve University Institutional Animal Care and Use Committee (IACUC). Housing and procedures complied with the Guide for the Care and Use of Laboratory Animals and the American Veterinary Medical Association guidelines on euthanasia. *SCoR2/Akr1a1* refers to the same gene and protein (*Zhou et al., 2019*), so '*Akr1a1*$^{-/-}$' and '*SCoR2*$^{-/-}$' in this versus other publications refer to an identical mouse line. *SCoR2*$^{+/-}$ (*Akr1a1*$^{+/-}$) mice were made by Deltagen, Inc as described previously (*Zhou et al., 2019*). Briefly, exon 2 was replaced with a LacZ/NEO cassette to disrupt in-frame translation, and mice were genotyped by PCR using primers forward: 5'-GCAGAGATTCAACAAGTCTCCCCTC-3'; mutant reverse: 5'-GGGCCAGCTCATTCCTCCCACTCAT-3'; and wild-type reverse: 5'-AGCTAAGGCTCCGAGCAGTGCTAAC-3'. The mice used in these studies were bred using *SCoR2*$^{-/-}$ male mice mated with *SCoR2*$^{+/-}$ female mice (since *SCoR2*$^{-/-}$ female mice do not breed efficiently) to generate *SCoR2*$^{-/-}$ offspring used for the experiments described here. *SCoR2*$^{+/+}$ mice were the offspring of *SCoR2*$^{+/-}$ male and female mice and were occasionally backcrossed during the generation of the data in this manuscript. *SCoR2*$^{-/-}$ mice were fed with AIN-93M diet supplemented with 1% L-ascorbic acid (vitamin C) (Research Diets, Cat# D16021012, custom diet) to correct for the inability of *Akr1a1/SCoR2*$^{-/-}$ mice to synthesize L-ascorbic acid, as

described previously (*Lai et al., 2017*). SCoR2$^{+/+}$ mice were fed with a control AIN-93M diet (Research Diets, Cat# D10012M).

## Myocardial I/R injury

Healthy male SCoR2$^{+/+}$ and SCoR2$^{-/-}$ mice (C57BL/6J background) aged 9–16 weeks were used for surgery. The female heart differs from the male heart in post-MI survival and cardiac function (*Shioura et al., 2008*; *Kam et al., 2004*; *Bridgman et al., 2005*; *Brower et al., 2003*; *Korte et al., 2005*) and is more resilient following injury (*Pullen et al., 2020*), making protection from tissue injury in female SCoR2$^{-/-}$ mice difficult to discern. Thus, distinct biology underlies the cardiac response to injury in female mice, and SCoR2$^{-/-}$-mediated cardioprotection in female mice is deserving of a separate study. Myocardial I/R injury was carried out as described previously (*Zhang et al., 2016*). Briefly, mice were anesthetized, intubated, and actively ventilated (respiration rate = ~133 /min and tidal volume = ~200 µl). The thoracic cavity was accessed through the left, third intercostal space, and the pericardium was resected to visualize the LAD artery. A ligature of 7-0 silk suture was placed around the LAD and tightened to fully occlude arterial flow as verified by ischemic change in coloration of the anterior wall and by EKG tracings (ST-segment elevation). Following 30 min of ischemia, reperfusion was initiated by untying the suture. Following the closure of the chest, pneumothorax was resolved by blunt needle aspiration. Mice were given 0.9% saline i.p. (0.5 ml/25 g body weight) and were recovered under temperature support. Following 1–4 hr reperfusion, mice for organ harvesting were anesthetized, whole blood harvested from IVC, spun and plasma collected, and LV, RV, and lung tissue harvested and snap-frozen in liquid nitrogen for biochemical analysis. Mice harvested for tissue following 1 hr reperfusion were excluded from survival analysis (*Figure 1G*). Mice destined for infarct size evaluation following 24 hr reperfusion were anesthetized, the chest was opened, and a catheter was inserted into the transverse aorta. The right atrium was opened and 0.5 ml of 0.9% saline was administered through the aortic cannula to exsanguinate the heart. The LAD was then occluded with the in-dwelling suture, and 0.2 ml of Evans blue (2% wt/vol) was injected into the LAD via the ascending aorta before the heart was extracted. The isolated heart was then frozen and sliced into segments 1 mm thick, and slices were placed in 1% TTC (2,3,5-triphenyltetrazolium chloride) in saline at 37°C for 20 min. TTC-stained slices were fixed in 10% formalin for 20 min before photographic documentation. Images were analyzed as described previously (*Zhang et al., 2016*), with Image Pro Plus to evaluate area-at-risk (red), area of necrosis (white), and total LV area.

## Echocardiography

Following 24 hr reperfusion, mice were anesthetized with 2,2,2-tribromoethanol (0.25 mg/g body weight, IP), and transthoracic echocardiography was performed using a Vevo 770 High-Resolution Imaging System equipped with an RMV-707B 30-MHz probe (VisualSonics). Standard M-mode sampling was used through the left ventricular short axis at the midpapillary level. Ejection fraction, fractional shortening, LVID-s, and other parameters were determined using the system's software.

## Troponin-1 and LDH measurement

Mouse cardiac troponin-1 and LDH were measured using troponin-1 ELISA (cat#: KT-470; Kamiya) and LDH activity assay (cat#: MAK066-1KT; Sigma) kits, respectively, according to the manufacturer's instructions.

## Terminal uridine deoxynucleotidyl transferase dUTP nick end labeling (TUNEL assay)

Heart tissue from SCoR2+/+and SCoR2-/- mice post-MI was harvested, fixed in 4% paraformaldehyde overnight, switched to PBS, and sectioned with 10 µm thickness onto microscope slides in preparation for TUNEL assay. Slides were baked at 60°C for 75 min, deparaffinized in two changes of xylene, rehydrated in graded ethanols (100%, 95%, 70%), and washed in PBS. Slides were pretreated with IHC Select Proteinase K (Chemicon International; cat# 21627) at final concentration of 20 µg/ml for 10 min at 25°C, then washed twice in dH$_2$O. Endogenous peroxidase activity was quenched with Peroxidazed 1 (BioCare Medical; cat# PX968G) for 8 min at 25°C, followed by 2x washes in dH$_2$O. Excess liquid was removed and 900 µl equilibration buffer (Chemicon Intl ApopTag Peroxidase In Situ Apoptosis Detection Kit; cat# S7100) applied for 10 s at 25°C. Excess liquid tapped off, and TdT enzyme applied

(ApopTag kit, 80% reaction buffer + 20% TdT enzyme) for 1 hr incubation at 37°C in a humidified chamber. TdT enzyme activity stopped with stop/wash buffer, agitated for 15 s, incubated for 10 min at 25°C, and washed 3x in PBS. Anti-digoxigenin peroxidase conjugate (ApopTag kit) was applied and incubated for 30 min at 25°C in a humidified chamber, followed by four washes in PBS. Two drops DAB chromogen added to 2.0 ml DAB substrate to form Betazoid DAB (BioCare; cat# BDB2004L), and slides incubated in Betazoid DAB for 5 min in the dark at 25°C, followed by 3 washes in dH$_2$O and 5 min incubation in dH$_2$O. Slides were then counterstained with CAT Hematoxylin (BioCare; cat# CATHE-M) for 30 s, rinsed in dH$_2$O, dipped 10 x in PBS, rinsed in tap water, air dried overnight, dipped in xylene for 10 s, and coverslip applied with a resinous medium. Slides were imaged using brightfield 10x on a confocal microscope, TUNEL$^+$ nuclei quantified and analyzed by GraphPad Prism.

### *S*-Nitrosothiol resin assisted capture

SNORAC (*Forrester et al., 2009*) was carried out as described previously (*Zhou et al., 2019*). Mouse heart samples were mechanically homogenized in lysis buffer containing 100 mM HEPES/1 mM EDTA/100 µM neocuproine (HEN), 50 mM NaCl, 0.1% (vol/vol) Nonidet P-40, the thiol-blocking agent 0.2% *S*-methylmethanethiosulfonate (MMTS), 1 mM PMSF and protease inhibitor (Roche). After centrifugation (10,000 × *g*, 4°C, 20 min, x2), SDS and MMTS were added to the supernatant to 2.5% and 0.2%, respectively, and incubated at 50° C for 20 min. Proteins were precipitated with –20°C acetone, and redissolved in 1 ml of HEN/1% SDS, repeated, and the final pellets were resuspended in HEN/1% SDS and protein concentrations determined using the BCA method. Total lysates (typically 1–2 mg) were incubated with freshly prepared 50 mM ascorbate and 100 µl thiopropyl-Sepharose (50% slurry; made in-house as described previously; *Seth et al., 2023*) and rotated end-over-end in the dark for 3 hr. The bound SNO-proteins were sequentially washed with HEN/1% SDS and 10% HEN/0.1% SDS; SNO-proteins were then eluted with 10% HEN/1% SDS/10% β-mercaptoethanol and analyzed by SDS/PAGE and immunoblotting with antibodies targeting proteins of interest.

### Protein extraction and digestion (SCoR2 nitrosoproteome analysis)

For the SCoR2 nitrosoproteome analysis (SNORAC coupled to LC–MS/MS), SNO-proteins from heart lysates from *N* = 4 mice for each condition were bound to thiopropyl-Sepharose beads, eluted using 2X Laemmli buffer with 10% β-mercaptoethanol, and provided to IQ Proteomics (Framingham, MA). Proteins were extracted by methanol–chloroform precipitation and digested with 0.5 µg of LysC for 12 hr at room temperature followed by 1 µg of trypsin (Promega) in 100 mM EPPS buffer (pH 8.0) for 4 hr at 37°C. A miscleavage rate check was performed on two random samples to ensure efficient digestion of proteins. Each of the eight tryptic peptide samples was labeled with 120 µg of Tandem Mass Tag (TMT10; Pierce) isobaric reagents for 2 hr at room temperature. A label efficiency check was performed by pooling 1 µl from each sample within a single plex to ensure at least 98% labeling of all N-termini and lysine residues. All samples were quenched with hydroxylamine (0.5%), acidified with trifluoroacetic acid (2%), pooled, and dried by vacuum centrifugation. Pooled TMT10 labeled peptides were reconstituted in 0.1% TFA and subjected to orthogonal basic-pH reverse phase fractionation on a 2.1 × 50 mm column packed with 1.8 µm ZORBAX Extend-C18 material (Agilent) and equilibrated with buffer A (5% acetonitrile in 10 mM ammonium bicarbonate, pH 8). Peptides were fractionated utilizing a 7-min linear gradient from 5% to 50% buffer B (90% acetonitrile in 10 mM ammonium bicarbonate, pH 8) at a flow rate of 0.3 ml/min. A total of six fractions were collected and consolidated into three fractions and vacuum dried. The peptides were reconstituted in 0.2% FA and desalted with Empore-C18 (3 M) in-house packed StageTips prior to analysis by mass spectrometry (MS).

### MS analysis and data search (SCoR2 nitrosoproteome analysis)

All mass spectra were acquired on an Orbitrap Fusion Lumos coupled to an EASY nanoLC-1200 (Thermo Fisher) liquid chromatography system by IQ Proteomics (Framingham, MA). Approximately 2 µg of peptides were loaded on a 75-µm capillary column packed in-house with Sepax GP-C18 resin (1.8 µm, 150 Å, Sepax) to a final length of 35 cm. Peptides were separated using a 120-min linear gradient from 10% to 40% acetonitrile in 0.1% formic acid. The mass spectrometer was operated in a data-dependent mode. The scan sequence began with FTMS1 spectra (resolution = 120,000; mass range of 350–1400 *m/z*; max injection time of 50 ms; AGC target of 1e6; dynamic exclusion for 60 s with a ±10 ppm window). The 10 most intense precursors were selected within a 2-s cycle

and fragmented via collisional-induced dissociation in the ion trap (normalized collision energy (NCE) = 35; max injection time = 35 ms; isolation window of 0.7 Da; AGC target of 1e4). Following the ITMS2 acquisition, a real-time search algorithm was employed to score each peptide and only trigger synchronous-precursor-selection (SPS) MS3 quantitative spectra for high confidence scoring peptides as determined by a linear discriminant approach. Following MS2 acquisition, an SPS MS3 method was enabled to select eight MS2 product ions for high energy collisional-induced dissociation with analysis in the Orbitrap (NCE = 55 for TMT10; resolution = 50,000; max injection time = 86 ms; AGC target of 1.4e5; isolation window at 1.2 Da for +2 $m/z$, 1.0 Da for +3 $m/z$, or 0.8 Da for +4 to +6 $m/z$). All mass spectra were converted to mzXML using a modified version of ReAdW.exe. MS/MS spectra were searched against a concatenated 2018 mouse Uniprot protein database containing common contaminants (forward + reverse sequences) using the SEQUEST algorithm (*Eng et al., 1994*). Database search criteria are as follows: fully tryptic with two missed cleavages; a precursor mass tolerance of 50 ppm and a fragment ion tolerance of 1 Da; oxidation of methionine (+15.9949 Da) was set as differential modifications. Static modifications were carboxyamidomethylation of cysteines (+57.0215 Da) and TMT on lysines and N-termini of peptides (modification mass for TMT10 is +229.1629 Da). Peptide-spectrum matches were filtered using linear discriminant analysis (*Huttlin et al., 2010*) and adjusted to a 1% peptide false discovery rate (FDR) (*Elias and Gygi, 2007*) and collapsed further to a final 1.0% protein-level FDR. Proteins were quantified by summing the total reporter intensities across all matching PSMs.

## LC–MS/MS analysis and data search (SCoR2 interactome)

For the SCoR2 interactome analysis, hearts isolated from $N$ = 4 wild-type mice were mechanically homogenized in non-denaturing lysis buffer (20 mM Tris-HCl (pH 8.0), 137 mM NaCl, 1% NP-40, 2 mM EDTA), clarified, and incubated with 10 µg of anti-SCoR2/AKR1A1 antibody (15054–1-AP, Proteintech) for 2 hr at room temperature. The sample was then incubated with 0.25 mg pre-washed protein A/G magnetic beads (cat# 88802; Thermo Scientific) for 1 hr at room temperature while mixing and placed into a magnet for washing steps. Proteins were eluted from beads with 0.1 M glycine (pH 2.0) for 10 min at room temperature with agitation, magnetically separated, sample collected, neutralized with 15 µl alkaline buffer (Tris, pH 8.0), and sent to the CWRU Proteomics and Small Molecule MS Core facility. Proteins were reduced with 10 mM dithiothreitol for 1 hr at 37°C, followed by alkylation with 25 mM iodoacetamide for 30 min in the dark. After reduction/alkylation of proteins, dual digestion was performed with 0.3 µg of lysyl endopeptidase on a dry bath shaker for 1 hr at 37°C followed by the addition of 0.3 µg of trypsin with incubation overnight at 37°C. After digestion, samples were filtered with 0.22 µM spin filters to remove any particulates before MS analysis (Costar Spin-X; Sigma-Aldrich). Peptides were identified using data-dependent MS acquisition.

Samples were analyzed on an Orbitrap Eclipse mass spectrometer (Thermo Electron, San Jose, CA) equipped with a Waters nanoACQUITY LC system (Waters, Taunton, MA). Peptides were desalted in a trap column (180 µm × 20 mm, packed with C18 Symmetry, 5 µm, 100 Å; Waters) and subsequently resolved on a reversed-phase column 75 µm × 250 mm nanocolumn, packed with C18 BEH130, 1.7 µm, 130 Å (Waters). Liquid chromatography was carried out at ambient temperature at a flow rate of 300 nl/min using a gradient mixture of 0.1% formic acid in water (solvent A) and 0.1% formic acid in acetonitrile (solvent B). The gradient used ranged from 1% to 60% solvent B over 90 min. Nanospray was conducted in a positive ion mode with a voltage of 2.4 kV. A full scan was obtained for eluted peptides in the range of 375–1500 atomic mass units followed by 25 data-dependent MS/MS scans.

MS/MS spectra were generated by collision-induced dissociation of the peptide ions at NCE of 35% to generate a series of b- and y-ions as major fragments. A 1-hr wash step was performed between each sample.

LC-MS/MS results were analyzed in Mascot (version 2.7.0) (Matrix Science, London, United Kingdom). The database used was *Mus musculus* UniProt (17,089 proteins). Search settings were as follows: trypsin enzyme specificity; mass accuracy window for precursor ion, 10.0 ppm; mass accuracy window for fragment ions, 0.60 Da, and 1 missed cleavage. For the interactome experiment, carbamidomethyl of cysteine was specified in Mascot as a fixed modification and oxidation of methionine was specified in Mascot as a variable modification.

## LC/MS-based untargeted metabolite profiling (metabolomic analysis)

Metabolite extraction and ratiometric profiling utilized acetonitrile (ACN), isopropanol (IPA), and methanol (MeOH) that were purchased from Fisher Scientific. High-purity Millipore filtered and deionized water (ddH$_2$O; >18 MOhm) was used in extraction media and chromatography mobile phases. OmniTrace glacial acetic acid and ammonium hydroxide were obtained from EMD Chemicals. Ammonium acetate, ammonium formate, and all other chemicals and standards were obtained from Sigma-Aldrich in the best available grade.

The left ventricle was isolated from mice post-sham, 1 hr I/R, or 4 hr I/R, $N = 5$ each condition and snap frozen in liquid nitrogen. Once received, tissue samples were washed with ice-cold PBS, followed by metabolite extraction using −70°C 80:20 methanol:water (LC/MS grade methanol, Fisher Scientific). The tissue–methanol mixture was subjected to bead-beating for 45 s using a Tissuelyser cell disrupter (QIAGEN). Extracts were centrifuged for 5 min at 18,500 × $g$ to pellet insoluble protein, and supernatants were transferred to clean tubes. The extraction procedure was repeated two additional times, and all three supernatants were pooled, dried in a Vacufuge (Eppendorf), and stored at −80°C until analysis. The methanol-insoluble protein pellet was solubilized in 0.2 M NaOH at 95°C for 20 min and protein was quantified using a Bio-Rad DC assay. On the day of metabolite analysis, dried cell extracts were reconstituted in 70% acetonitrile at a relative protein concentration of 4 μg/ml, and 4 μl of this reconstituted extract was injected for LC/MS-based targeted and untargeted metabolite profiling.

Tissue extracts were analyzed by LC/MS as described previously (*Chen et al., 2018*), using a platform comprised of an Agilent Model 1290 Infinity II liquid chromatography system coupled to an Agilent 6550 iFunnel time-of-flight MS analyzer. Chromatography of metabolites utilized aqueous normal phase chromatography on a Diamond Hydride column (Microsolv, cat# 70000-15D-2, 4 μm, 2.1 mm ID × 150 mm length, 100 A). Mobile phases consisted of (A) 50% isopropanol, containing 0.025% acetic acid, and (B) 90% acetonitrile containing 5 mM ammonium acetate. To eliminate the interference by metal ions on chromatographic peak integrity and electrospray ionization, EDTA was added to the mobile phase at a final concentration of 5 μM. The following gradient was applied: 0–1.0 min, 99% B; 1.0–15.0 min, to 20% B; 15.0–29.0, 0% B; 29.1–37 min, 99% B. Raw data were analyzed using MassHunter Profinder 8.0 and MassProfiler Professional (MPP) 15.1 software (Agilent). Student's $t$-tests ($p < 0.05$) were performed to identify significant differences between groups.

Peripheral blood was isolated fresh from murine IV post-sham, 1 hr I/R, or 4 hr I/R, $N = 5$ each condition, into BD microtainer (cat#365967; BD) tubes for blood separation after 30 min room-temperature incubation followed by centrifugation at 1500 × $g$ for 5 min. Plasma was then snap-frozen and sent for LC/MS analysis. Plasma metabolites were extracted by the addition of 1 part plasma to 20 parts 70% acetonitrile in ddH$_2$O (vol:vol). The mixture was briefly vortexed and then centrifuged for 5 min at 16,000 × $g$ to pellet precipitated proteins. An aliquot of the resulting extract (3 μl) was subjected to LC/MS-based untargeted metabolite profiling in both positive and negative ion modes.

## Metabolite structure specification

To ascertain the identities of differentially expressed metabolites ($p < 0.05$), LC/MS data were searched against an in-house annotated personal metabolite database created using MassHunter PCDL manager 8.0 (Agilent), based on monoisotopic neutral masses (<5 ppm mass accuracy) and chromatographic retention times. A molecular formula generator (MFG) algorithm in MPP was used to generate and score empirical molecular formulae, based on a weighted consideration of monoisotopic mass accuracy, isotope abundance ratios, and spacing between isotope peaks. A tentative compound ID was assigned when the PCDL database and MFG scores concurred for a given candidate molecule. Tentatively assigned molecules were verified based on a match of LC retention times and/or MS/MS fragmentation spectra for pure molecule standards contained in a growing in-house metabolite database.

## Bioinformatics

All studies were done in at least triplicate, unless otherwise noted. Studies using mice utilized at least 3 mice per condition/treatment. All metabolomic studies used 5 mice per group. Co-IP mass spectrometry data (CWRU) were obtained as unique spectrum count, and SNORAC mass spectrometry data (IQ Proteomics) were obtained as normalized summed signal/noise values for each identified

protein. In the interactome study, proteins with an average of ≥1 unique spectrum count across three mouse heart samples were identified as interacting proteins and included in the 'interactome' list. In the SNORAC/MS nitrosoproteome study, all SNO-proteins identified in SCoR2$^{-/-}$ mouse heart with values ≥1.2-fold change relative to those identified in SCoR2$^{+/+}$ mouse heart, in at least two out of four experimental sets, were included in the final 'SNO-ome' list.

## Western blot analysis

Proteins were extracted from cells or tissues and subjected to 4–20% Criterion Precast Midi Protein Gel electrophoresis. Blotted membranes were incubated overnight at 4°C with primary antibodies, washed with PBS +0.1% Tween-20, then incubated with HRP-conjugated secondary antibody (anti-mouse or anti-rabbit IgG (Promega)) for 1 hr followed by chemiluminescent detection (ECL (GE Healthcare)). Antibodies employed in Western blotting included: rabbit polyclonal anti-AKR1A1/SCoR2 (15054-1-AP, Proteintech Group), rabbit monoclonal anti-PKM2 (D78A4, Cell Signaling), mouse monoclonal anti-p97 (10R-P104A, Fitzgerald), rabbit polyclonal anti-BDH1 (15417-1-AP, Proteintech Group). Coomassie blots used Imperial Protein Stain (cat#24615; Thermo Scientific) with 1 hr staining followed by 3 hr washing. Quantification of Western blots was carried out with ImageJ (NIH).

## Assay of NADPH-dependent SCoR2 activity with SNO-CoA or carbohydrates as substrate

The NADPH-dependent SNO-CoA reductase activity of recombinant SCoR2 was determined spectrophotometrically as described previously. Briefly, the assays were performed in 50 mM phosphate buffer (pH 7.0; containing 0.1 mM EDTA and DTPA) with 186 nM SCoR2, 100 µM NADPH and substrate: 100 µM SNO-CoA as positive control or 1 mM of specific test carbohydrates (L-(+)-arabinose, D-(-)-ribose, D-(-)-erythrose, D-(+)-glucose, D-(-)-fructose, D-(+)-galactose, sucrose, D-ribulose, and D-xylulose). Reactions were initiated by the addition of recombinant SCoR2 and allowed to proceed for 1 min while NADPH consumption was measured by spectrophotometer.

## NADPH-Glo assay

NADPH/NADP$^+$ assay was performed with NADP/NADPH-Glo Assay kit (Promega), as per the manufacturer's instructions. Mouse heart samples (2 µg/ml) were mechanically homogenized in EBC lysis buffer containing protease inhibitor cocktail (Roche), clarified, plated in triplicate, and manufacturer's protocol followed. Luminescence was recorded using Promega GloMax Discover instrument.

## cGMP measurement

GMP ELISA was performed using Cyclic GMP Complete kit (Enzo; cat#ADI-900-164) as per the manufacturer's instructions. Mouse heart samples (post-I/R with 1 hr reperfusion) were homogenized in 10 volumes of 0.1 M HCl, centrifuged at 600 × $g$ for 10 min, and supernatant diluted 1:20 and run directly in the assay. Assay results in pmol cGMP/ml were normalized between samples by dividing by protein concentration as determined by BCA assay, and results displayed as pmol cGMP/mg total protein.

## Human subjects

De-identified human post-mortem myocardial specimens with basic clinical data (sample type, sex, age, and race) were collected by the Duke Human Heart Repository under a non-human subjects research exemption granted by the Duke University Medical Center Institutional Review Board (# Pro00005621). A similar exemption was granted by the Case Western Reserve University IRB (study#: STUDY20230090) for experiments conducted at CWRU with this human tissue. Patient population characteristics are presented in *Supplementary file 4*.

## Mouse subjects

All experimental procedures were approved by the Institutional Animal Care and Use Committee of Case Western Reserve University School of Medicine (protocol 2013-0091) and were conducted in accordance with the NIH *Guide for the Care and Use of Laboratory Animals* (National Academies Press, 2011). All potentially painful procedures were conducted under a surgical plane of anesthesia (typically ketamine/xylazine 100/10 mg/kg), and mice were sacrificed by methods compliant with the American Veterinary Medical Association (AVMA) *Guidelines for Euthanasia of Animals* (2013 edition).

## Statistical analysis

Data are presented as mean ± SD except where noted, with median and range reported in supplementary materials (*Supplementary file 3*). Parametric and non-parametric tests of statistical significance, as appropriate and indicated within each figure legend, were used to test for strain and treatment effects. Parametric tests were used if normality was satisfied, and non-parametric tests were used if normality was not satisfied or could not be confirmed due to $n < 5$ independent replicates. Analysis was performed using GraphPad Prism software (v9). Under all conditions, p-values <0.05 were taken to indicate a statistically significant difference between groups.

## Acknowledgements

The authors acknowledge the assistance of Kathleen Lundberg at the Proteomics and Small Molecule Mass Spectrometry Core and of Adam Kresak and Jennifer Mikulan at the Tissue Resource Core at Case Western Reserve University, Isobel Taylor for assisting with metabolomics data acquisition at Weill Cornell Graduate School of Medical Sciences, the staff at the Human Heart Repository at Duke University, and the IQ Proteomics team in Framingham, MA. This work was supported in part by: National Institutes of Health grant P01HL158507 (JSS). National Institutes of Health grant R01HL126900 (JSS). National Institutes of Health grant RO1DK119506 (JSS). National Institutes of Health grant RO1DK128347 (JSS). American Heart Association-Allen grant 19PABH134580006 (JSS). National Institutes of Health grant R01HL157151 (WJK and JSS) National Institutes of Health grant RO1AR076029 (QC). National Institutes of Health grant RO1NS131322 (SSG). National Institutes of Health MSTP grant 2T32GM007250-44 (ZWG).

## Additional information

### Competing interests

Jonathan S Stamler: Reviewing editor, eLife. The other authors declare that no competing interests exist.

### Funding

| Funder | Grant reference number | Author |
|---|---|---|
| National Heart Lung and Blood Institute | HL158507 | Jonathan S Stamler |
| National Heart Lung and Blood Institute | HL126900 | Jonathan S Stamler |
| National Heart Lung and Blood Institute | HL157151 | Walter J Koch Jonathan S Stamler |
| National Institute of Diabetes and Digestive and Kidney Diseases | DK119506 | Jonathan S Stamler |
| National Institute of Diabetes and Digestive and Kidney Diseases | DK128347 | Jonathan S Stamler |
| American Heart Association | 19PABH134580006 | Jonathan S Stamler |
| National Institute of Arthritis and Musculoskeletal and Skin Diseases | AR076029 | Qiuying Chen |
| National Institute of Neurological Disorders and Stroke | NS131322 | Steven S Gross |
| National Institute of General Medical Sciences | GM007250-44 | Zachary W Grimmett |

| Funder | Grant reference number | Author |
|---|---|---|

The funders had no role in study design, data collection, and interpretation, or the decision to submit the work for publication.

## Author contributions

Zachary W Grimmett, Conceptualization, Formal analysis, Investigation, Methodology, Writing – original draft, Writing – review and editing; Rongli Zhang, Formal analysis, Investigation; Hua-Lin Zhou, Investigation, Methodology; Qiuying Chen, Dawson Miller, Formal analysis, Investigation, Methodology; Zhaoxia Qian, Resources; Justin Lin, Riti Kalra, Investigation; Steven S Gross, Walter J Koch, Richard T Premont, Supervision, Writing – review and editing; Jonathan S Stamler, Conceptualization, Supervision, Funding acquisition, Project administration

## Author ORCIDs

Zachary W Grimmett ![ORCID] https://orcid.org/0000-0003-3820-2049
Qiuying Chen ![ORCID] https://orcid.org/0000-0001-5909-3959
Richard T Premont ![ORCID] https://orcid.org/0000-0002-8053-5026
Jonathan S Stamler ![ORCID] https://orcid.org/0000-0002-6866-1572

## Ethics

De-identified human post-mortem myocardial specimens with basic clinical data (sample type, sex, age, and race) were collected by the Duke Human Heart Repository (Duke University IRB# Pro00005621) and obtained in the de-identified state. Experiments at CWRU involving these samples (study#: STUDY20230090) were determined by the CWRU IRB to be categorized as research not involving human subjects and therefore not requiring separate CWRU IRB review and approval.
All experimental procedures were approved by the Institutional Animal Care and Use Committee of Case Western Reserve University School of Medicine and were conducted in accordance with the NIH Guide for the Care and Use of Laboratory Animals (National Academies Press, 2011). All potentially painful procedures were conducted under a surgical plane of anesthesia (typically ketamine/xylazine 100/10 mg/kg), and mice were sacrificed by methods compliant with the American Veterinary Medical Association (AVMA) Guidelines for Euthanasia of Animals (2013 edition).

Reviewer #1 (Public review): https://doi.org/10.7554/eLife.106601.3.sa1
Reviewer #2 (Public review): https://doi.org/10.7554/eLife.106601.3.sa2
Reviewer #3 (Public review): https://doi.org/10.7554/eLife.106601.3.sa3
Author response https://doi.org/10.7554/eLife.106601.3.sa4

# Additional files

## Supplementary files

Supplementary file 1. SCoR2-dependent cardiac *S*-nitrosoproteome and SCoR2 interactome.

Supplementary file 2. SCoR2-dependent cardiac and plasma metabolome.

Supplementary file 3. Statistical values for all quantitative experimental results.

Supplementary file 4. Population characteristics of patients from whom heart samples were assessed.

MDAR checklist

## Data availability

All data are available in the main text or in the supplementary materials.

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

# Appendix 1

## Appendix 1—key resources table

| Reagent type (species) or resource | Designation | Source or reference | Identifiers | Additional information |
|---|---|---|---|---|
| Strain, strain background (mouse, male and female) | Akr1a1$^{tm1Dgen}$, mixed 129x C57BL/6 | Deltagen; *Zhou et al., 2019* | | Referred to as SCoR2-KO |
| Genetic reagent (*E. coli*) | OneShot Omnima competent cells | Invitrogen | C8540-03 | |
| Genetic reagent (*E. coli*) | BL21-CodonPlus competent cells | Agilent | 230240 | |
| Cell line (human) | HEK293 | ATCC | CRL-1573 | |
| Transfected construct (human) | BDH1 cDNA in pDONR223 | DNAsu | Clone: HsCD00352477 | |
| Transfected construct (human) | BDH1 cDNA in pcDNA-DEST40 | This study | | Mammalian expression, V5 epitope tag |
| Transfected construct (human) | BDH1 C115S cDNA in pcDNA-DEST40 | This study | | Mammalian expression, C115S mutant, V5 epitope tag |
| Biological sample (human) | Post-mortem myocardium | Duke Human Heart Repository | | |
| Antibody | Anti-AKR1A1/SCoR2 (rabbit polyclonal) | Proteintech | 15054-1-AP | 10 µg for IP; 1:1000 for blotting |
| Antibody | Anti-PKM2 (rabbit monoclonal) | Cell Signaling | D78A4 | 1:1000 for blotting |
| Antibody | Anti-p97 (mouse monoclonal) | Fitzgerald | 10R-P104A | 1:1000 for blotting |
| Antibody | Anti-BDH1 (rabbit polyclonal) | Proteintech | 15417-1-AP | 1:1000 for blotting |
| Recombinant DNA reagent | AKR1A1 cDNA in pET21b | *Anand et al., 2014* | | Bacterial expression, CT 6xHis tag for purification |
| Sequence-based reagent | forward: 5'-CGTCCAGCTCAATGTCTCCAGCAGCGAAGAGG reverse: 5'-CCTCTTCGCTGCTGGAGACATTGAGCTGGACG | This study | | PCR primers for human SCoR2 C115S mutagenesis |
| Sequence-based reagent | forward: 5'-GCAGAGATTCAACAAGTCTCCCCTC mutant reverse: 5'-GGGCCAGCTCATTCCTCCCACTCAT wild-type reverse: 5'-AGCTAAGGCTCCGAGCAGTGCTAAC | *Zhou et al., 2019* | | PCR primers for mouse SCoR2 (Akr1a1) genotyping |
| Peptide, recombinant protein | SCoR2/AKR1A1-6xHis | *Anand et al., 2014* | | Bacterial expression, CT 6xHis tag for purification |
| Commercial assay or kit | QuikChange II site-directed mutagenesis kit | Agilent | 200523 | |
| Commercial assay or kit | Troponin-1 ELISA kit | Kamiya | KT-470 | |
| Commercial assay or kit | LDH activity assay | Sigma | MAK066-1KT | |
| Commercial assay or kit | NADP/NADPH-Glo assay | Promega | G9071 | |
| Commercial assay or kit | Cyclic GMP Complete ELISA kit | Enzo | ADI-900-164 | |

*Appendix 1 Continued on next page*

*Appendix 1 Continued*

| Reagent type (species) or resource | Designation | Source or reference | Identifiers | Additional information |
|---|---|---|---|---|
| Chemical compound, drug | Thiopropyl-Sepharose | *Seth et al., 2023* | | |
| Software, algorithm | GraphPad Prism statistics software (v9) | https://www.graphpad.com/ | | |
| Other | | | | |

