## [Editor Report · eLife Assessment]

This study provides new and interesting findings that SCoR2 acts as a denitrosylase to control cardioprotective metabolic reprogramming and prevent injury following ischemia/reperfusion. The **compelling** evidence is supported by a novel multi-omics approach, but questions remain regarding the stability and human relevance of BDH1 as well as the sufficiency of SCoR2. Overall, the work will be of interest to cardiovascular researchers and provides **valuable** information to the field, though some mechanistic aspects require further clarification.

---

## [Referee Report · Reviewer #1 (Public review)]

Summary:

This study shows a novel role for SCoR2 in regulating metabolic pathways in the heart to prevent injury following ischemia/reperfusion. It combines a new multi-omics method to determine SCoR2 mediated metabolic pathways in the heart. This paper would be of interest to cardiovascular researchers working on cardioprotective strategies following ischemic injury in the heart.

Strengths:

(1) Use of SCoR2KO mice subjected to I/R injury.

(2) Identification of multiple metabolic pathways in the heart by a novel multi-omics approach.

Comments on revisions:

Authors have addressed all concerns raised in the previous round of review. Substantial modifications have been made in response to those concerns. There are no further comments.

---

## [Referee Report · Reviewer #2 (Public review)]

Summary:

This manuscript addresses the gap in knowledge related to the cardiac function of the S-denitrosylase SNO-CoA Reductase 2 (SCoR2; product of the Akr1a1 gene). Genetic variants in SCoR2 have been linked to cardiovascular disease, yet its exact role in heart remains unclear. This paper demonstrates that mice deficient in SCoR2 show significant protection in a myocardial infarction (MI) model. SCoR2 influenced ketolytic energy production, antioxidant levels, and polyol balance through the S-nitrosylation of crucial metabolic regulators.

Strengths:

Addresses a well-defined gap in knowledge related to the cardiac function of SNO-CoA Reductase 2. Besides the in-depth case for this specific player, the manuscripts sheds more light on the links between S-nytrosylation and metabolic reprogramming in heart.

Rigorous proof of requirement through the combination of gene knockout and in vivo myocardial ischemia/reperfusion

Identification of precise Cys residue for SNO-modification of BDH1 as SCoR2 target in cardiac ketolysis

Weaknesses:

The experiments with BDH1 stability were performed in mutant 293 cells. Was there a difference in BDH1 stability in myocardial tissue or primary cardiomyocytes from SCoR2-null vs -WT mice? Same question extends to PKM2.

In the absence of tracing experiments, the cross-sectional changes in ketolysis, glycolysis or polyol intermediates presented in Figures 4 and 5 are suggestive at best. This needs to be stressed while describing and interpreting these results.

The findings from human samples with ischemic and non-ischemic cardiomyopathy do not seem immediately or linearly in line with each other and with the model proposed from the KO mice. While the correlation holds up in the non-ischemic cardiomyopathy (increased SNO-BDH1, SNO-PKM2 with decreased SCoR2 expression), how do the Authors explain the decreased SNO-BDH1 with preserved SCoR2 expression in ischemic cardiomyopathy? This seems counterintuitive as activation of ketolysis is a quite established myocardial response to the ischemic stress. It may help the overall message clarity to focus the human data part on only NICM patients.

(partially linked to the point above) an important proof that is lacking at present is the proof of sufficiency for SCoR2 in S-Nytrosylation of targets and cardiac remodeling. Does SCoR2 overexpression in heart or isolated cardiomyocytes reduce S-nitrosylation of BDH1 and other targets, undermining heart function at baseline or under stress?

Comments on revisions:

Some of my points have been addressed. However, the points related to (1) BDH1 stability effect in cardiomyocytes; (2) human relevance of SNO-BDH1; (3) SCoR2 sufficiency remain unclear. That said, this manuscript will provide useful information to the field as such.

---

## [Referee Report · Reviewer #3 (Public review)]

Summary:

This manuscript demonstrates that mice lacking the denitrosylase enzyme SCoR2/AKR1A1 demonstrate a robust cardioprotection resulting from reprogramming of multiple metabolic pathways, revealing

widespread, coordinated metabolic regulation by SCoR2.

Strengths:

The extensive experimental evidence provided the use of the knockout model

Weaknesses:

No direct evidence for the underlying mechanism.

The mouse model used is not a tissue-specific knock-out.

---

## [Author Response]

The following is the authors’ response to the original reviews.

**Reviewer #1 (Public review):**
Summary:This study shows a novel role for SCoR2 in regulating metabolic pathways in the heart to prevent injury following ischemia/reperfusion. It combines a new multi-omics method to determine SCoR2 mediated metabolic pathways in the heart. This paper would be of interest to cardiovascular researchers working on cardioprotective strategies following ischemic injury in the heart.Strengths:(1) Use of SCoR2KO mice subjected to I/R injury.(2) Identification of multiple metabolic pathways in the heart by a novel multi-omics approach.We thank the Reviewer for the positive review of our manuscript.Weaknesses:(1) Use of a global SCoR2KO mice is a limitation since the effects in the heart can be a combination of global loss of SCoR2.(2) Lack of a cell type specific effect.

We agree that global KOs limit the cell type-specific mechanistic conclusions that can be drawn. Global knockouts are nonetheless informative in their own right and serve to identify phenotypes worthy of further study.

**Reviewer #2 (Public review):**
Summary:This manuscript addresses the gap in knowledge related to the cardiac function of the S-denitrosylase SNOCoA Reductase 2 (SCoR2; product of the Akr1a1 gene). Genetic variants in SCoR2 have been linked to cardiovascular disease, yet their exact role in the heart remains unclear. This paper demonstrates that mice deficient in SCoR2 show significant protection in a myocardial infarction (MI) model. SCoR2 influenced ketolytic energy production, antioxidant levels, and polyol balance through the S-nitrosylation of crucial metabolic regulators.Strengths:(1) Addresses a well-defined gap in knowledge related to the cardiac function of SNO-CoA Reductase 2. Besides the in-depth case for this specific player, the manuscript sheds more light on the links between Snitrosylation and metabolic reprogramming in the heart.(2) Rigorous proof of requirement through the combination of gene knockout and in vivo myocardial ischemia/reperfusion.(3) Identification of precise Cys residue for SNO-modification of BDH1 as SCoR2 target in cardiac ketolysis

We thank the Reviewer for their kind words.

Weaknesses:(1) The experiments with BDH1 stability were performed in mutant 293 cells. Was there a difference in BDH1 stability in myocardial tissue or primary cardiomyocytes from SCoR2-null vs -WT mice? The same question extends to PKM2.

We have not assessed BDH1 stability directly in cardiomyocytes. However, S-nitrosylation increased BDH1 stability in HEK293 cells, and BDH1 expression was increased in (injured) hearts of SCoR2KO mice, together with increased SNO-BDH1.

For PKM2, there is a wealth of published evidence from us and others that S-nitrosylation does not regulate protein stability but rather inhibits tetramerization required for full activity.

(2) In the absence of tracing experiments, the cross-sectional changes in ketolysis, glycolysis, or polyol intermediates presented in Figures 4 and 5 are suggestive at best. This needs to be stressed while describing and interpreting these results.

We now acknowledge this limitation in the ‘Limitations’ section of the manuscript and in edits made to the text.

(3) The findings from human samples with ischemic and non-ischemic cardiomyopathy do not seem immediately or linearly in line with each other and with the model proposed from the KO mice. While the correlation holds up in the non-ischemic cardiomyopathy (increased SNO-BDH1, SNO-PKM2 with decreased SCoR2 expression), how do the authors explain the decreased SNO-BDH1 with preserved SCoR2 expression in ischemic cardiomyopathy? This seems counterintuitive as activation of ketolysis is a quite established myocardial response to ischemic stress. It may help the overall message clarity to focus the human data part on only NICM patients.

We find it interesting and important that SNO-BDH1 is readily detected in human heart tissue and its level is correlated to disease state. Our findings suggest conservation of this mechanism in human heart failure. However, we caution against drawing further conclusions related to NICM or ICM. Our animal model (based on a single time point) cannot faithfully recapitulate patients with chronic heart disease or differences between NICM and ICM.

(4) This is partially linked to the point above. An important proof that is lacking at present is the proof of sufficiency for SCoR2 in S-nitrosylation of targets and cardiac remodeling. Does SCoR2 overexpression in the heart or isolated cardiomyocytes reduce S-nitrosylation of BDH1 and other targets, undermining heart function at baseline or under stress?

The Reviewer proposes to test the effect of SCoR2 overexpression on cardioprotection. This is an interesting experiment for future study with the following caveats. First, it presupposes that native expression of SCoR2 is insufficient to control basal steady state S-nitrosylation of SNO-BDH1 and SNO-PKM2 (this does not seem to be the case). Second, overexpressed SCoR2 may be mislocalized within cells or associated with unnatural targets. Thank you.

**Reviewer #3 (Public review):**
Summary:This manuscript demonstrates that mice lacking the denitrosylase enzyme SCoR2/AKR1A1 demonstrate a robust cardioprotection resulting from reprogramming of multiple metabolic pathways, revealing widespread, coordinated metabolic regulation by SCoR2.Strengths:(1) The extensive experimental evidence.(2) The use of the knockout model.

We thank the Reviewer for identifying strengths in our work.

Weaknesses:(1) The connection of direct evidence for the mechanism.

We believe we have identified a novel mechanism for cardioprotection entailing coordinate reprogramming of multiple metabolic pathways and suggesting a widescale role for SCoR2 in metabolic regulation. This is the key message we convey. While genetic dissection of individual pathways may be worthwhile, these investigations will have their own limitations.

(2) The mouse model used is not tissue-specific.

Please see our response to Reviewer 1, above.

**Reviewer #1 (Recommendations for the authors):**
In the study, titled "The denitrosylase SCoR2 controls cardioprotective metabolic reprogramming", Grimmett ZW et al., describe a role for SNO-CoA Reductase 2 (SCoR2) in promoting cardioprotection via metabolic reprogramming in the heart after I/R injury. Authors show that loss SCoR2 coordinates multiple metabolic pathways to limit infarct size. Overall, the hypothesis is interesting, however there are some limitations as described below:(1) It is unclear whether SCoR2 mice are global or cardiomyocyte specific.

We apologize for any confusion. These are global SCoR2^-/-^ mice. This is now stated in the Results when first identifying the strain, as well as in the Methods.

(2) Can the authors clarify how divergent metabolic pathways such as Ketone oxidation, glycolysis, PPP and polyol metabolism work downstream of SCoR2 to impact cardioprotection in mice with I/R.

The metabolic pathways of ketone oxidation, glycolysis, PPP and polyols appear to converge to support ischemic cardioprotection in SCoR2^-/-^ mice, as depicted in the model shown in Fig. 5L. Subsequent to SNO-PKM2 blockade of flux through glycolysis (detailed in this manuscript and in Zhou et al, 2019, PMID: 30487609, as well as by others), substrates of ketolysis and glycolysis are funneled into the PPP, producing the antioxidant NADPH and energy precursor phosphocreatine, which are well-known to be cardioprotective. This occurs more readily in SCoR2^-/-^ mice due to elevated SNO-BDH1 (detailed in this manuscript).

Polyols, thought to be products of the PPP carbohydrate intermediates arabinose, ribulose, xylulose (among others), have recently been shown to be harmful to cardiovascular health in humans. These polyols are uniformly downregulated in SCoR2^-/-^ mice. We suggest this is likely the result of S-nitrosylation of SCoR2-substrate enzymes that form polyols (SCoR2/Akr1a1 is unable to directly reduce carbohydrates to their corresponding polyols). Regulation of endogenous polyol production in humans is a new concept and the mechanisms whereby these compounds increase risk of cardiac events are a subject of active investigation. This is detailed in the final paragraph of both the Results and Discussion sections, and in Fig. 5L.

(3) The only functional outcome of SCoR2 loss in echocardiography and measurements for apoptosis. However, it would be important to determine whether the cardioprotective effect persists. It seems cardiac function was recorded 24hours post injury and whether the benefit remains till later time point such as 2 or 4 weeks is not shown. Without this time point, loss of SCoR2 only leads to an acute increment in function.

Loss of SCoR2 reduced post-MI mortality at 4 hr; cardiac functional changes (plus troponin, LDH, and apoptosis) were studied in surviving animals at 24 hr post-MI. Cardiac response to acute injury and to chronic injury (weeks post-MI) are not the same metabolically. This is well elucidated in the literature and exemplified by the role of PKM2, which is protective in the chronic response to MI (28 days post-MI; PMID: 32078387), but implicated in injury at shorter timepoints post-MI (PMID: 33288902, 28964797). All that said, functional changes at 2-4 weeks will be important to determine in the future, as the Reviewer indicates.

**Reviewer #2 (Recommendations for the authors):**
(1) The last paragraph of the Results section should be divided into the statement related to Table S2 in the Results section, and the rest of the paragraph should be put somewhere in the Discussion.

Thank you for this suggestion, which we have taken.

(2) The number of mice alive/dead should be reported in the histogram in Figure 1G.

Done.

(3) A concise Graphical Abstract will be useful to grasp the overall logic and message of the manuscript from the beginning.

We thank you for this suggestion and have added a graphical abstract to the manuscript.

**Reviewer #3 (Recommendations for the authors):**
I would suggest having more evidence on the effect of metabolic reprogramming on which cell type. The use of a global knockout is a major limitation, and probably some in vitro experiments with shRNA knockdown in endothelial cells and fibroblasts would provide more insights.

The reviewer suggests one direction for future study. We identify a novel mechanism for cardioprotection entailing coordinate reprogramming of multiple metabolic pathways and suggesting a widescale role for SCoR2 in metabolic regulation. This is the message we wish to convey. The role of cardiomyocytes vs contributing cell types is a thoughtful direction for future study. Thank you.

**Editor's additional comment:**
The editors wish to highlight a critical issue concerning the characterization of the SCoR2−/− mice employed in this study.In the Methods section (page 20), the manuscript states that "SCoR2+/− mice were made by Deltagen, Inc as described previously (33)." However, reference 33 does not describe SCoR2−/− mice; instead, it refers to other genetically modified strains, including Akr1a1+/−, eNOS−/−, and PKM2−/− mice, with no mention of a SCoR2-targeted model.The editors fully acknowledge that the authors may be using the term "SCoR2" as a functional synonym for Akr1a1, based on its described role as a mammalian homologue of yeast SCoR. If this is the case, such equivalence should be explicitly stated in the manuscript to prevent potential confusion. Moreover, considering that the genetic deletion of Akr1a1 (i.e., SCoR2) underlies the key mechanistic findings presented, it is essential that the manuscript include a clear and comprehensive description of the generation and validation of the mouse model used.We therefore ask the authors to (1) clarify the nomenclature and relationship between "SCoR2" and Akr1a1, and (2) provide full details on the generation of the knockout mice, including the targeting strategy and the genotyping procedures. This information is necessary not only to ensure transparency and reproducibility but also to allow readers to fully appreciate the biological relevance of the findings.

Thank you for identifying this inconsistency. We have adjusted the manuscript text accordingly to clearly state that SCoR2 is a functional name for the product of the Akr1a1 gene and that these SCoR2^-/-^ mice are the same as Akr1a1^-/-^ mice described in Ref 33. We have augmented the Methods text to describe the generation and genotyping of these SCoR2/Akr1a1 knockout mice.